# Structural host-virus interactome profiling of intact infected cells

Boris Bogdanow [1,2,6] ✉, Lars Mühlberg [1,3,6], Iris Gruska [4], Barbara Vetter[4], Julia Ruta [1], Arne Elofsson [5], Lüder Wiebusch [4] ✉ & Fan Liu [1,3] ✉

Virus-host protein-protein interactions (PPIs) are fundamental to viral infections, yet high-resolution identification of their structural and molecular determinants within the native context of intact infected cells has remained an unsolved challenge. Here, we provide detailed insights into the structural interactome of herpes simplex virus 1-infected human cells by combining in-cell cross-linking mass spectrometry with the selective enrichment of newly synthesized viral proteins. In productively infected cells, we obtain 739 PPIs based on 6,194 cross-links found across intracellular compartments and at the intact host endomembrane system. These structural host-virus interactome profiling (SHVIP) data resolve PPIs to the protein domain level and augment AlphaFold-based structural modeling, facilitating detailed predictions of PPI sites within structured and intrinsically disordered regions. Importantly, SHVIP captures parts of the virus-host PPI space that are elusive to traditional interaction proteomics approaches. Validation by molecular genetics confirms that these new SHVIP identifications are genuine virus-host PPIs occurring in the complex environment of intact infected cells.

During an infection, viruses interact with the host's proteome, initiating processes that lead to viral genome replication, encapsidation, and the release of new progeny. These processes are facilitated by protein-protein interactions (PPIs), forming the physical and functional interface between the viral proteome and the host's cellular proteome.

Gaining a systematic understanding of the structural context and molecular determinants of virus-host PPIs remains a major challenge. While structural biology techniques have provided insight into the assembly of various viral protein complexes[1–5], the number of solved virus-host structures outside of receptor and antibody binding is scarce. Furthermore, viral proteins frequently contain short linear motifs (SLiMs) in their intrinsically disordered regions (IDRs) for physical association with host proteins[6] but viral SLiMs are typically predicted based on the amino acid composition in the primary structure, without directly considering co-evolutionary and structural

relationships[7]. Both these limitations are potentially addressable with recently developed structure prediction algorithms, such as Alpha-Fold2 (AF2)-based AF-Multimer and AlphaFold3 (AF3)[8,9]. However, AF has been developed mainly based on structural data of multimeric complexes within one organism, with co-evolutionary constraints likely different in host-pathogen scenarios[10,11]. Therefore, the utility of AF tools for predicting virus-host dimers at the interactome scale remains to be established. More generally, the need for orthogonal experimental evidence to validate and improve in silico-generated models remains critical[12].

A second challenge pertains to the molecular environment in which virus-host PPIs are studied. Viruses induce significant cellular changes, affecting signaling pathways[13], energy metabolism[14], protein subcellular localization[15], and gene expression[16,17]. Viruses also alter the spatial organization and microenvironment of the cell, forming

[1]Research group "Structural Interactomics", Leibniz Forschungsinstitut für Molekulare Pharmakologie, Berlin, Germany. [2]Charité - Universitätsmedizin Berlin, corporate member of Freie Universität Berlin and Humboldt-Universität zu Berlin, Institut für Virologie, Berlin, Germany. [3]Charité - Universitätsmedizin Berlin, Berlin, Germany. [4]Labor für Pädiatrische Molekularbiologie, Department of Pediatric Oncology and Hematology, Charité - Universitätsmedizin Berlin, Berlin, Germany. [5]Stockholm Bioinformatics Center, Stockholm University, Stockholm, Sweden. [6]These authors contributed equally: Boris Bogdanow, Lars Mühlberg. ✉e-mail: boris.bogdanow@charite.de; lueder.wiebusch@charite.de; fliu@fmp-berlin.de

membrane-less organelles via liquid-liquid phase separation (LLPS)[18–21] and remodeling the host membrane system to create fragile viral macromolecular assemblies[22]. Studying virus-host PPIs in the context of these processes requires an intact cellular environment. However, a systematic survey of virus-host PPIs in intact, natively infected cells is lacking. The reasons for this paucity are illustrated by the following three examples. First, affinity purification mass spectrometry (AP-MS), the most widely used approach for systematic virus-host PPI profiling[23–33], is not applicable to intact infected cells, cannot discriminate between direct and indirect interactors, and does not provide information on the interaction contact site. Second, virus-host PPI screens based on proximity biotinylation (e.g., BioID and APEX) require genetically engineered and ectopically expressed viral proteins[34,35], which may behave differently than during an actual infection. In addition, proximity biotinylation, akin to AP-MS, is unable to reveal direct interactors and interaction sites. Third, cryo electron tomography can visualize virus–host interactions at molecular resolution in native environments[36], but fails to provide systematic insights across the cell and cannot reveal the identity of interacting proteins in an untargeted fashion.

These challenges can, in principle, be addressed by cross-linking mass spectrometry (XL-MS)[37]. XL-MS captures PPIs with the help of small organic molecules (cross-linkers) that can covalently connect pairs of specific protein residue side chains, provided that they are within reach of the cross-linker. Considering that each cross-linker has a defined maximum length, these residue-to-residue connections (cross-links), which are identified by MS, represent distance constraints that provide insights into protein conformations (through cross-links within the same protein = intra-links), PPIs (through cross-links between different proteins = inter-links), and protein interfaces (through the location of the inter-linked residues). Importantly, XL-MS is applicable to intact wild-type cells[38,39]. However, the sensitivity of in-cell XL-MS is rather limited, meaning that the more abundant host-host PPIs may mask virus-host PPIs.

Here, we characterize the identity, structural determinants, and spatial context of hundreds of virus-host PPIs in intact cells that are productively infected with herpes simplex virus type 1 (HSV-1), a medically relevant human herpesvirus with a complex proteome[40]. To achieve this, we developed *Structural Host Virus Interactome Profiling (SHVIP)*, which utilizes the above-mentioned benefits of proteome-wide XL-MS and overcomes its sensitivity issues by combining XL-MS with selective bio-orthogonal labeling of viral proteins. The resulting HSV-1 interactome comprises known and novel PPIs supported by thousands of cross-links. Using these data as guidance for AF modeling allows resolving interaction interfaces in structured regions and predicting SLiMs within IDRs of viral proteins. By integrating our SHVIP results with affinity purification and molecular genetics, we provide the structural and molecular determinants of dozens of virus-host PPIs.

## Results

### Capturing the viral interactome in intact HSV-1 infected cells
In order to harness the advantages of XL-MS for the characterization of intact infected cells, we needed to enhance its sensitivity to viral proteins. Therefore, we decided to enrich viral proteins by capitalizing on the host shutoff, which is common in many viral infections[41]. During host shutoff, a large fraction of newly synthesized proteins in infected cells is of viral origin, which can be captured by metabolic pulse labeling with bio-orthogonal amino acids[42–45].

Our SHVIP strategy presented here combines XL-MS with bio-orthogonal labeling and involves four steps (Fig. 1a). First, cells are infected with a virus that is able to replicate efficiently in a given cell line. Second, once cells are committed to a replicative viral life cycle, L-homopropargylglycine (HPG) is added to label newly synthesized— and thus predominantly viral—proteins for several hours. Third, the

membrane-permeable cross-linker DSSO is added to the intact cells to covalently link lysine side chains from proteins that are in proximity with each other. Fourth, HPG-containing proteins are extracted using copper-catalyzed click chemistry together with their covalently linked interactors. Finally, the enriched proteins are digested, enriched for cross-linked peptides via strong cation exchange chromatography, and analyzed by MS.

We applied this methodology to HSV-1-infected human embryonic lung fibroblasts (HELFs). Fibroblasts support the HSV-1 lytic replication cycle, which is characterized by a temporal cascade of viral gene expression and the shutoff of host protein synthesis[46]. The host shutoff is achieved by a combination of transcriptional and post-transcriptional mechanisms that are installed within the first 3–6 h after infection by the concerted action of ICP4, ICP22, ICP27, and the viral endonuclease UL41 (also konwn as Vhs). After this early phase of infection, HSV-1, like all herpesviruses, has a nuclear stage of viral DNA replication and capsid assembly, and a cytoplasmic stage in which the outer tegument and membrane envelope layers of the newly forming virus particles are assembled. The peak of virus progeny production and release is typically reached around 24 h post-infection. Based on this timeline, we decided to label the cells with HPG from 7 to 24 h post-infection.

MS analysis of the linear (non-cross-linked) peptides in the enriched sample and an aliquot taken before HPG-enrichment (input) shows that HPG-enrichment reduces host protein abundance by a factor of 10 on average (Fig. 1b and Supplementary Data 1). Similarly, it increases the frequency of sequenced spectra of viral origin (Supplementary Fig. 1a), and expands the fraction of viral proteins contributing to the overall intensity from ~20% to ~75% (Supplementary Fig. 1b). The contribution of viral proteins to the total protein intensity from cross-linked samples was slightly lower (~60%) as additional cross-linked host proteins co-purified with the viral proteins. Importantly, SHVIP substantially increased the proportion of identified cross-links involving viral proteins (Fig. 1c). The highest increase was observed for cross-links between peptides originating from different protein sequences (inter-links), which is particularly valuable as inter-links give deeper insight into the viral PPI network. We evaluated two acquisition schemes, based on either MS2-MS3 (Experiment 1) or MS2-only (Experiment 2) fragmentation. Both the relative frequency and absolute numbers of PPIs involving viral proteins increased upon HPG enrichment (Supplementary Fig. 1c) in both experiments. Reproducibility between the two experiments (Supplementary Fig. 1d) was similar to previous XL-MS studies from complex samples[38]. Taken together, this demonstrates that SHVIP increases the sensitivity of viral interactome capture.[46]

### Spatial resolution of the virus-host interactome map
In order to provide a high-coverage structural interactome of the infected cell, we analyzed the combined enriched and input samples from two experiments involving different MS-acquisition schemes. We first filtered inter- and intra-links separately at a 1% residue pair-level false discovery rate (FDR). We then focused on the most relevant PPIs and generated a subnetwork of interactions among viral proteins and their direct host interaction partners (Fig. 2a and Supplementary Data 2), yielding 739 interactions (of which 441 involve viral proteins) at a 1% separate PPI-level FDR estimated by target and decoy counts (see Methods). These data cover 46 viral proteins, representing ~68% of the viral proteins detected by MS-analysis of linear (non-cross-linked) peptides) peptides from enriched samples (Supplementary Data 1). We observed cross-links for viral structural (e.g., tegument, glycoproteins) and non-structural proteins (e.g., replication machinery) as well as a few cross-links to the rigid nucleocapsid shell of the virus (Fig. 2b). Published focused biochemical studies confirm ~21.5 % of the detected PPIs from infected cells (Supplementary Data 2 and Supplementary Fig. 1e), suggesting that our network captures functionally relevant

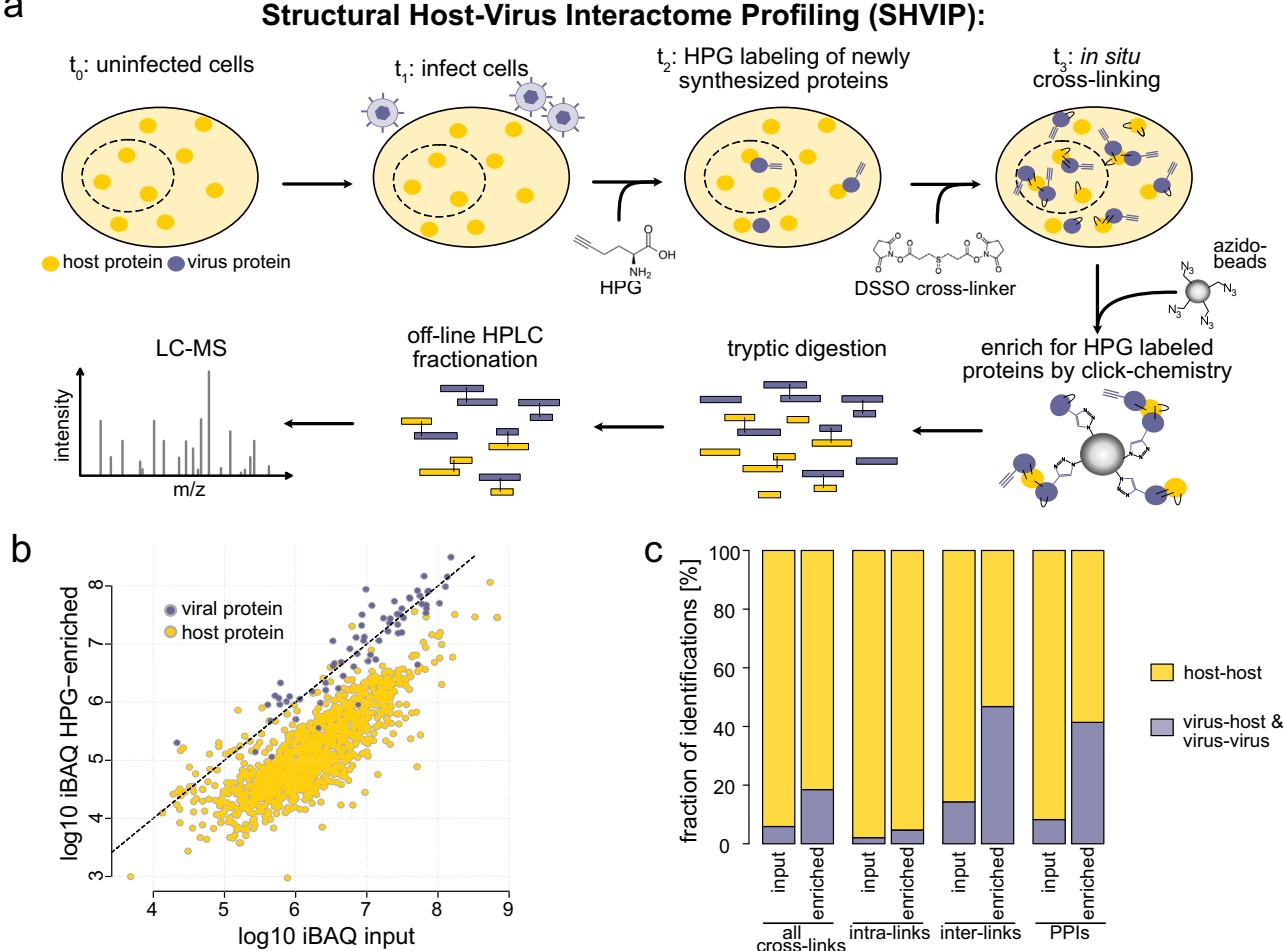

**Fig. 1 | SHVIP workflow and proof of concept. a** Workflow: cultured cells ($t_0$) are infected with a virus able to induce host shutoff ($t_1$). Once host shutoff is in place, newly synthesized proteins are labeled with L-HPG ($t_2$), and subsequently cross-linked with DSSO ($t_3$), and finally enriched using click chemistry. Proteins are digested into peptides, which are fractionated by off-line HPLC (in this case Strong cation exchange chromatography) and measured by LC-MS. **b** Abundance (measured by iBAQ) of host and viral proteins in non-cross-linked HPG-enriched and input samples. **c** Relative fraction of identifications involving viral proteins or only host proteins for different identification types in XL-MS data. Source data are provided.

interactions. Taken together, this indicates that SHVIP sensitively captures a large part of the available viral proteome.

To assess if SHVIP captures the virus interactome across the entire host cell, we performed unsupervised clustering based on centrality indices[47], dividing our network into six communities (Fig. 3a). A GO-enrichment analysis, based on cellular component annotations of host proteins, revealed distinctly enriched categories for these communities (Fig. 3b). While protein communities 4 and 5 enrich ribosomal proteins, community 6 contains many DNA binding proteins such as histones and other chromatin proteins (e.g., HMGB1, HMGB2, HMGN2). In two other communities (1 and 3), host proteins of the endomembrane system are enriched, such as proteins localizing to the endoplasmic reticulum (ER) lumen (e.g., B2M, CALR, P4HB, PDIA6, ERP44), ER-Golgi intermediate compartment or the Golgi apparatus (e.g., GALNT1, RAB6A, SCAMP2, SCAMP3). These data confirm that SHVIP can resolve virus-host PPIs across various cellular compartments.

We were particularly interested in the PPIs at the endomembrane system. During the late stage of herpesvirus infection, cellular membranes become reorganized to allow assembly and release of mature virions[22]. This is regulated by a complex network of PPIs, where membranes separate the involved proteins into a lumenal (of ER, trans-Golgi, or vesicles) or extra-lumenal (cytosol, nucleoplasm, or intra-

virion) population (Fig. 3c). While DSSO is able to pass the membrane, it is not able to cross-link proteins that are separated by membranes. Therefore, cross-links in this region allow evaluating whether SHVIP compromises the integrity of the cellular membranes. When evaluating cross-linking partners of viral envelope proteins at domain-level resolution (Fig. 3d), we found that almost all cross-links (99.6%) occurred between domains within either the lumenal or the extra-lumenal compartments, confirming that SHVIP captures interactions at a largely intact membrane system.

We mapped the domain-specific interaction partners onto HSV-1 transmembrane proteins, which validated 23 known PPIs (Fig. 3e, Supplementary Fig. 2a, and Supplementary Data 2). Specifically, we found viral tegument proteins localizing to the extra-lumenal side of the membrane and interacting with cytosolic tails of viral glycoproteins, consistent with their ascribed role in mediating secondary envelopment[48–50]. In addition, we observed many Rab GTPases, crucial regulators of endocytic and exocytic membrane trafficking, and related proteins at the cytosolic side. This includes not only RAB5[51], RAB1A/B[52], and RAB6[53] proteins with known proviral functions, but also RAB2A, which is yet uncharacterized in HSV-1 infection (Supplementary Fig. 2b).

At the lumenal side, we observed many well-known ER-resident proteins, such as chaperones (PDIA4, PDIA6, HSP90B1) and

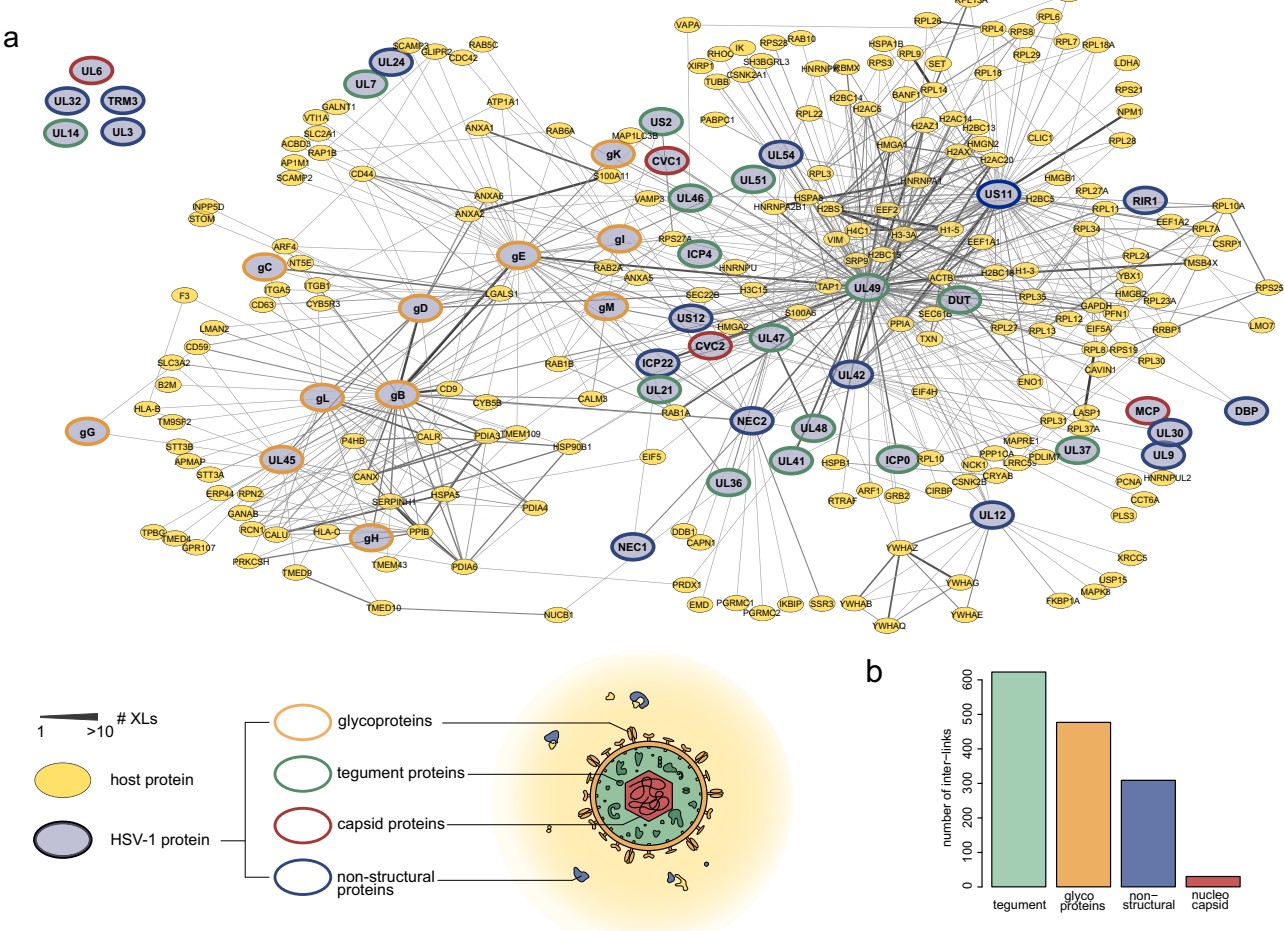

**Fig. 2 | The HSV-1-host protein interactome as captured by SHVIP. a** Virus-centric PPI network of HSV-1 infected cells. Host proteins (yellow) are only included when they are directly linked to viral proteins (blue). Data are filtered to 1 % separate cross-link FDR and reported at 1% inter-PPI FDR with 6194 cross-links and 254 proteins. Edge width scales with the number of identified cross-links between the two proteins. The outline color of the nodes indicates the different classes of viral proteins (glycoproteins in orange, tegument in green, capsid in red, and non-structural proteins in blue). **b** Inter-link (cross-link in between two different proteins) coverage for viral proteins from different classes. Source data are provided.

glycosylation factors (GALNT1, GANAB), interacting with the extra-virion domains of viral glycoproteins, consistent with ER-luminal processing of viral glycoproteins. Interestingly, the viral membrane proteins gB and UL45 physically associated with cellular proteins involved in antigen presentation, such as HLA-B, HLA-C, and B2M, potentially perturbing the cellular antigen presentation pathway[54]. In addition, glycoproteins gH/gL and gB, core components of the viral fusion machinery, were found cross-linked to alpha and beta subunits of integrins (ITGA5, ITGB1), which are known entry receptors for HSV-1[55] (Supplementary Fig. 2c). We also observed cross-links between gH/gL and gB as well as the fusion-regulatory protein UL45[56], supporting the idea of physical cross-talk among proteins of the HSV-1 fusion machinery[57].

Similar to these viral glycoproteins, we also found other viral proteins that cause substantial membrane remodeling, such as the nuclear egress complex (NEC), composed of NEC1 and NEC2. NEC mediates the envelopment of nucleocapsids at the inner nuclear membrane, followed by their delivery to the outer nuclear membrane by membrane fusion[58]. The in situ PPIs of this complex (Supplementary Fig. 2d) reveal both known and novel insight into this process. First, we observed known functionally relevant interaction partners such as ICP22[59] and EMD (also known as emerin)[60]. Second, we found the extra-lumenal domains of several viral glycoproteins associated with NEC2, previously discussed as functionally important for fusion at the outer

nuclear membrane[22]. Third, we discovered the association of five ER-membrane proteins to the NEC (IKBIP, SEC22B, SSR3, SEC61B, and LRRC59), suggesting a yet unknown role of ER proteins during nuclear egress. Overall, our data provides detailed, domain-level insights into virus-host interactomes at the endomembrane system, relevant for glycoprotein processing, membrane trafficking, adaptive immunity, egress, and assembly.

**Identifying PPIs that depend on the intact cellular environment**
Having established that SHVIP provides extensive and biologically relevant data, we aimed to identify which of those PPIs critically depend on the intact cellular environment. To this end, we compared our approach to AP-MS, which is the most widely employed method for virus-host PPI profiling[23–33] and requires cell lysis. We selected eight viral proteins from our network with different numbers of cross-linked interactors each. These included non-structural proteins (UL12, DBP, NEC2, and US11) as well as tegument proteins (UL47, UL48, UL49, and DUT) (see Methods). We tagged the proteins individually with an HA tag at their endogenous loci and performed anti-HA affinity purifications[27], enriching the bait with its interaction partners from infected cells. As controls, we used cells infected with the parental wildtype (WT) virus, not expressing any HA-tagged protein variants (Fig. 4a, Supplementary Fig. 3a–h, and Supplementary Data 3). On average, cross-linking partners of the bait show significantly higher

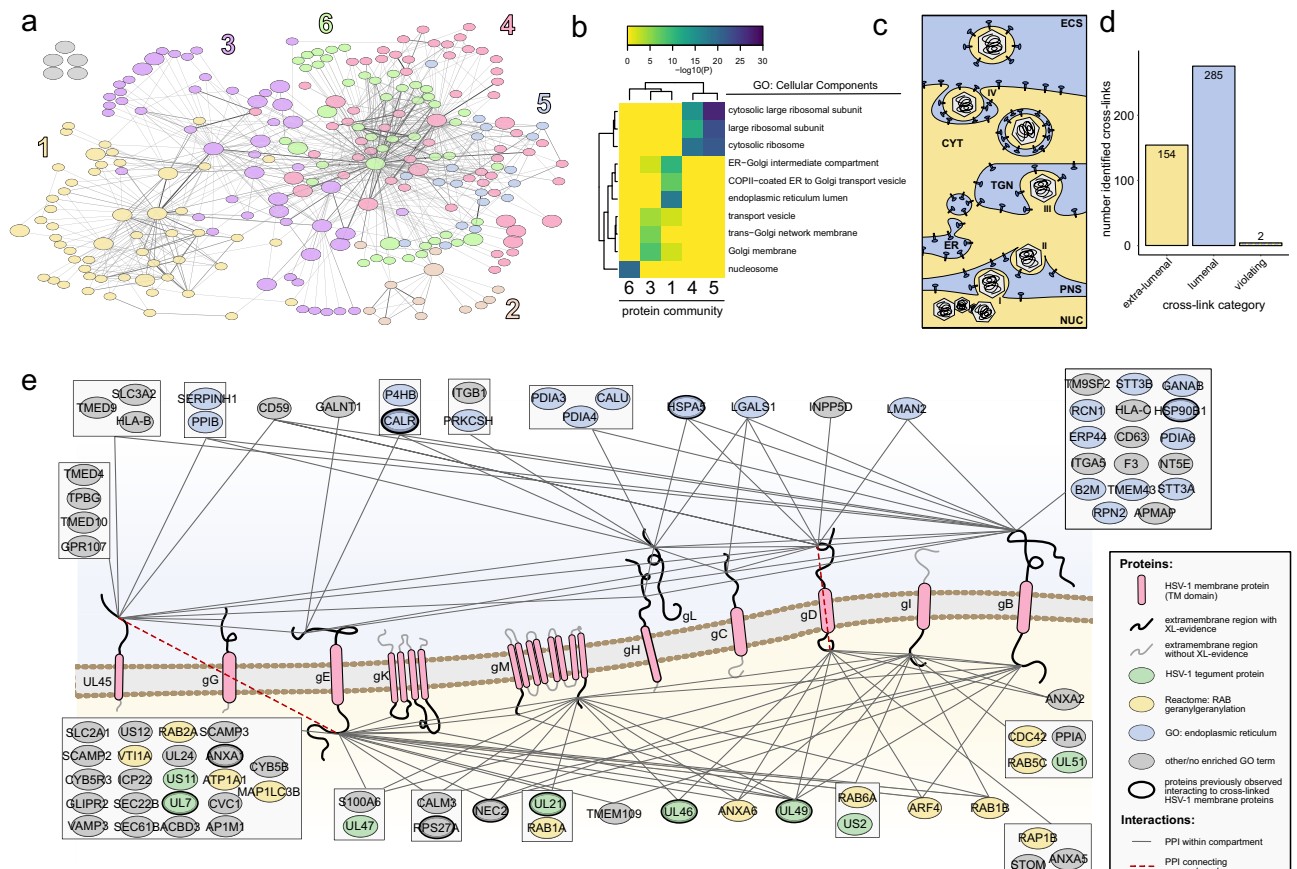

**Fig. 3 | The structural interactome of HSV-1 at the host endomembrane system.** **a** Unsupervised community clustering[47] of the structural interactome. Proteins colored according to the community numbers. The layout of the network is identical to the network layout in Fig. 2a. **b** GO-enrichment based on cellular component annotations of host proteins within five of the communities. Community 2 had too few host proteins to enrich any GO term. **c** Extra-lumenal (yellow, e.g., cytosol) and lumenal (blue, e.g., ER lumen) compartments in late stage HSV-1 infection (NUC nucleus, PNS perinuclear space, ER endoplasmatic reticulum, TGN trans-Golgi network, CYT cytosol, ECS extracellular space). **d** Number of cross-links connecting lumenal or extra-lumenal domains of viral transmembrane proteins and cross-linking partners with annotated domains. **e** Viral transmembrane proteins (red)

shown in a membrane with their cytosolic tails pointing down and lumenal domains pointing up. Interaction partners with their domain-specific cross-links to either lumenal or extra-lumenal domains are grouped when they share cross-links to the same transmembrane protein domains. Color coding of host interaction partners according to GO annotations in legend. Host proteins previously observed to interact with any of the cross-linked HSV-1 membrane proteins (see also Supplementary Data 2) are highlighted by bold outlines. Host transmembrane proteins with several domain-specific links to viral transmembrane proteins are provided in Supplementary Fig. 2a. Cross-links that are in violation with the membrane topology are highlighted by red dashed edges. Source data are provided.

enrichment in the corresponding anti-HA-bait precipitate than proteins with indirect or no cross-link connection to the bait (Fig. 4b, c). At the level of individual baits, we observed significant differences in 6 out of 8 cases (Supplementary Fig. 3i–p). In total, at a stringent AP-MS log2 fold-change cut-off of 4, 42 SHVIP interactors could be validated by AP-MS (Fig. 4d). At a less stringent cut-off at log2 of 2, the number of overlapping interactors increased to 61 and remained significant (Fig. 4e). By utilizing a sliding cut-off, we observed a progressive decrease in the significance of the overlap when interactions below this threshold were considered (Fig. 4f). Thus, SHVIP reported PPIs can be validated by AP-MS and both techniques retrieve a common set of PPIs.

Looking at individual protein examples shows that SHVIP and AP-MS capture overlapping as well as complementary parts of the interactome. For the viral proteins UL12 and UL47, almost all cross-linking partners were also found enriched in anti-HA precipitates (Fig. 4g). In the case of UL49, an abundant tegument protein with a wide variety of functions and interaction partners[61], we observed a more mixed picture when comparing AP-MS and SHVIP:

Several known UL49 interaction partners, such as casein kinase 2[62], UL41, UL48 (also known as VP16)[63], and SET (also known as TAF-I)[64],

were detected by XL-MS and were highly enriched in AP-MS (Fig. 4g, h). However, other biologically relevant UL49 interactors detected by XL-MS, e.g., ribosomal proteins (consistent with UL49's known polysome association[65]), displayed only intermediate AP-MS co-enrichment levels. Moreover, nucleosome-associated proteins were observed as the most frequent cross-linking partners of UL49, corroborating microscopy data[66–68] and in vitro experiments[67,68]. Surprisingly, they were not enriched in AP-MS, suggesting that the nucleosome-UL49 interaction is disrupted by cell lysis or the enrichment procedure. An explanation could be that these interactions are linked to UL49 undergoing LLPS with DNA[20], a concentration-dependent process that is disturbed by cell lysis. These results suggest that, while many PPIs are identified by both SHVIP and AP-MS, those depending on the in situ environment of the infected cell are uniquely captured by SHVIP.

## IDRs are enriched for virus-host contact sites
In addition to identifying interacting proteins in the infected cell, we aimed to provide structural insights into the cross-linked protein assemblies. This is possible as cross-linkers bridge a defined distance range, e.g., up to 35 Å for the DSSO cross-linker applied here. As such,

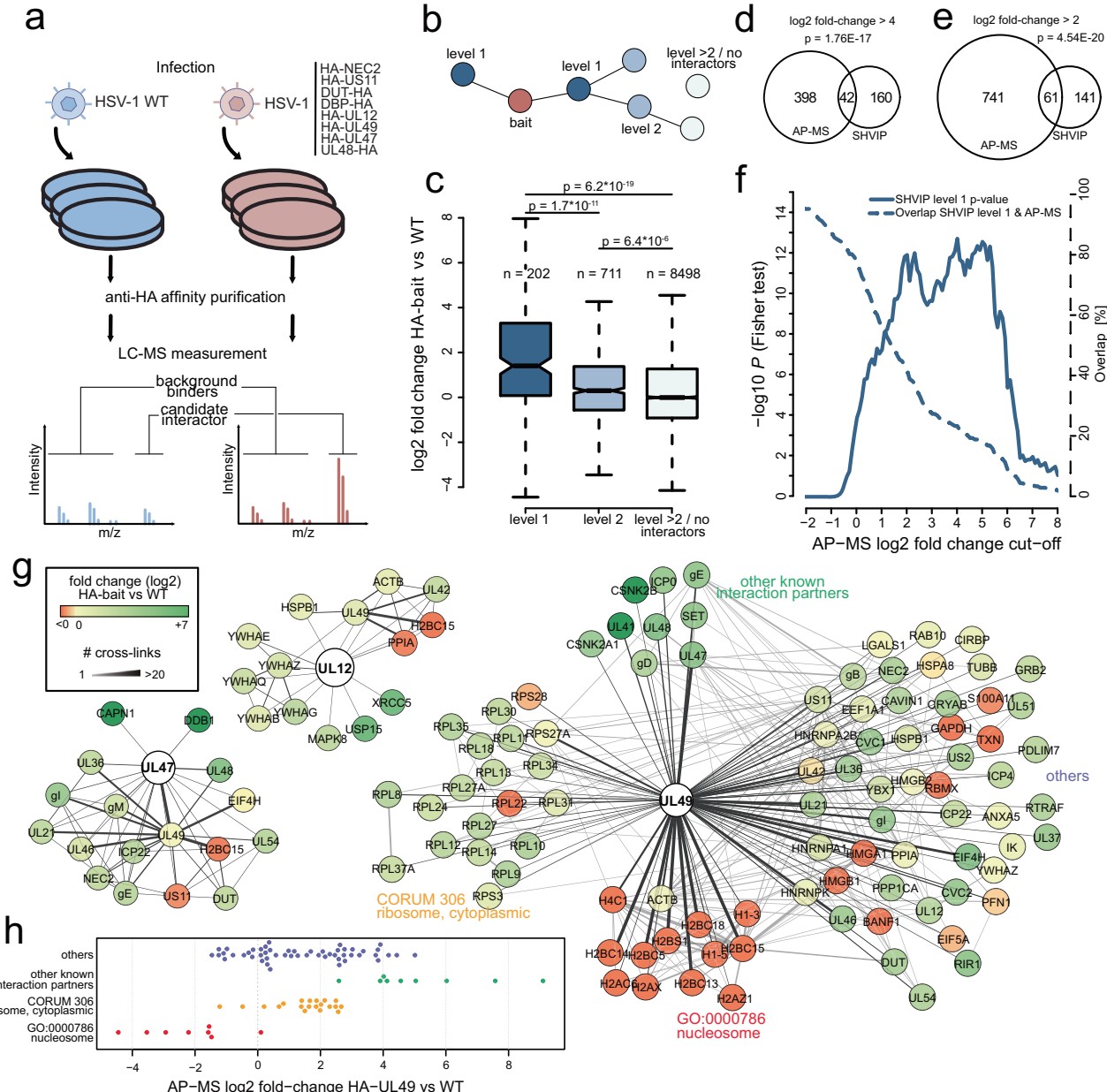

**Fig. 4 | SHVIP complements and extends AP-MS data. a** Workflow for AP-MS experiments. Transgenic HSV-1 viruses individually expressing the indicated HA-tagged protein variants were used to infect HELFs (*n* = 3 biological replicates), followed by anti-HA enrichment and LC-MS measurement. Purifications were compared to control experiments with HSV-1 wildtype (WT) not expressing any HA-tagged bait. The intensity ratio comparing HA-bait to control enrichments was computed for individual co-enriched proteins (Volcano plots for all baits are included in Supplementary Fig. 3). **b** For comparing to AP-MS data, proteins in the SHVIP interactome (see also Fig. 2a) were categorized into proteins cross-linked to the bait (level 1), proteins cross-linked to the bait via one other protein (level 2), or higher-order interactors and proteins not contained in the structural interactome (>level 2/no interactors). **c** Log2 fold-changes in the respective AP-MS experiment comparing the protein categories from (**b**). Boxes represent lower and upper quartiles with median marked as horizontal line. Whiskers represent 1.5 times interquartile range. Outliers (points outside the whiskers) removed for visibility.

*P* values are based on a two-sided Wilcoxon rank-sum test. Comparison of the interactome of the 8 selected viral proteins in SHVIP and AP-MS at a higher stringency (**d**) and lower stringency (**e**) log2 fold-change cut-off applied to AP-MS data. An additional *t* test *P* value cut-off (AP-MS *P* values, two-sided *t* tests without multiple hypothesis correction, *n* = 3 biological replicates) of 0.01 was applied to potential AP-MS interactors. The displayed *P* values indicate the statistical significance of the overlap between proteins identified via AP-MS and SHVIP and are based on one-sided Fisher's Exact test. **f** Comparing the overlap between SHVIP and AP-MS and its statistical significance using a sliding AP-MS fold-change cut-off. *P* values are based on one-sided Fisher's Exact test. **g** Representative depiction of SHVIP interactomes for selected proteins. Color coding of interacting nodes according to the co-enrichment ratios with the bait in AP-MS. **h** AP-MS co-enrichment levels of UL49 interactors found by SHVIP. Interactors are grouped into different biological categories. Source data are provided.

our data can support the structural characterization of heterodimers from HSV-1-infected cells.

Only five inter-links (3 virus-host or virus-virus PPIs) could be mapped onto experimental structures from the Protein Data Bank (PDB) (Fig. 5a), which is largely due to the scarce structural information

on multimeric HSV-1 complexes. Outside of receptor or antibody binding, only ~11 complexes have been solved, many of which only contain parts of the proteins. In order to extend insights into yet unsolved heterodimeric structures, we predicted all binary interactions in our dataset by AF-multimer[69] (Fig. 5b) and categorized model quality

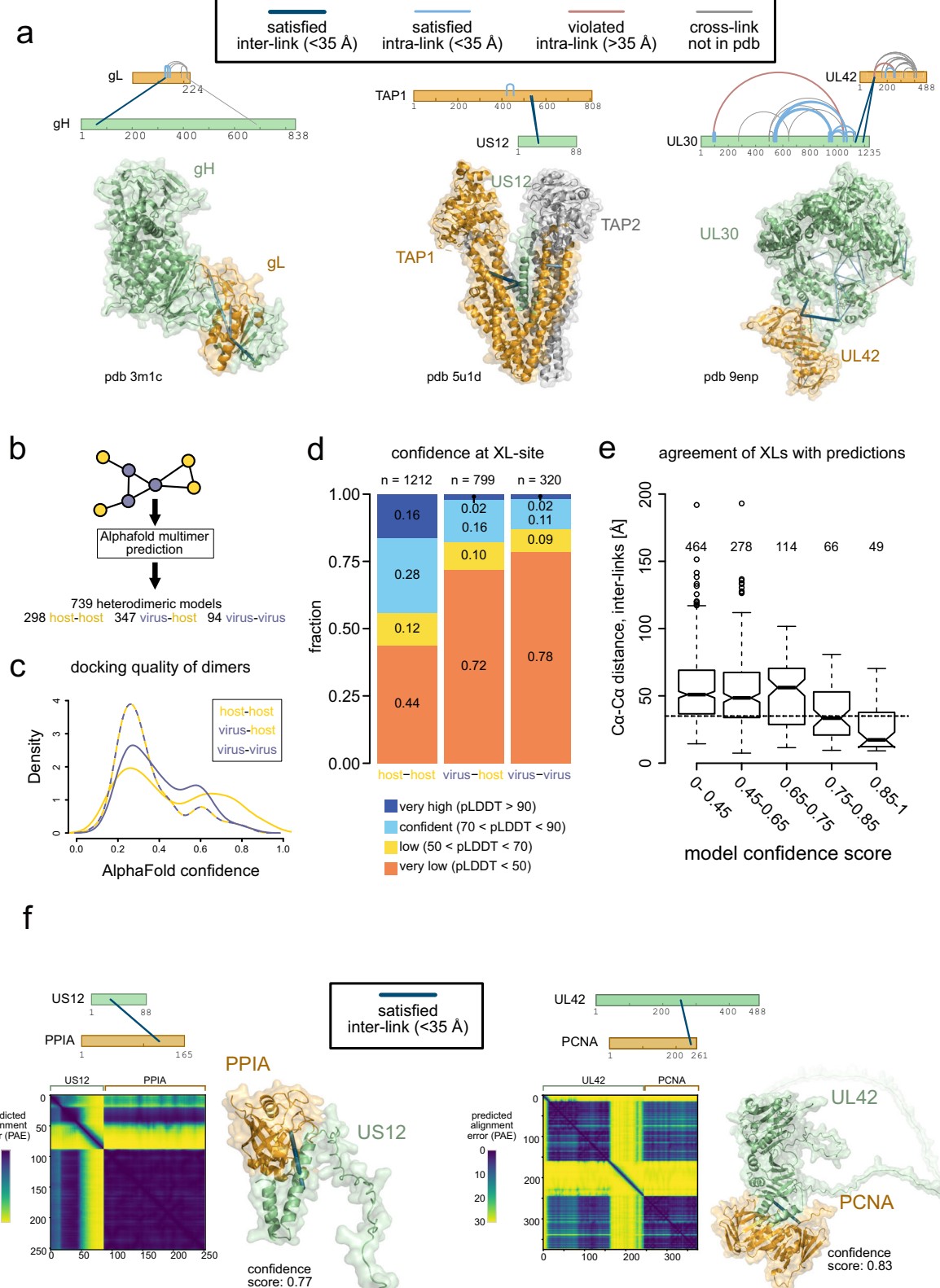

based on the confidence score derived from predicted template modeling score (pTM) and interface pTM (ipTM)[70]. In general, dimers between host proteins yielded predictions with better scores than those between viral proteins only or between viral and host proteins (Fig. 5c). This is consistent with the comparably low success rates of docking predictions for viral proteins[70], lower depth in the multiple sequence alignment (MSA) for viral proteins and lower expected co-

evolutionary signals captured in MSA between two organisms (host and virus). In addition, when comparing AF-Multimer models to cross-links, we noticed that inter-links involving viral proteins occurred more frequently in poorly predicted regions (pLDDT <50), likely IDRs, than those between host proteins (Fig. 5d). Thus, physical contacts within ordered and disordered regions shape the structural interactome of HSV-1, with the majority of them located in IDRs.

**Fig. 5 | Structural insight into virus-host PPIs through SHVIP and AF-Multimer.**
**a** Cross-links mapped onto heterodimers with experimentally solved structures.
Proteins shown as bars[109] or high-resolution experimental structures (gH-gL, pdb
3m1c, US12-TAP1/2, pdb 5u1d, and UL30-UL42, pdb 9enp). Note that gH-gL is an
experimental structure obtained from HSV-2. Cross-links highlighted gray could
not be mapped to the experimental structures because of missing coordinates. One
intra-link in UL30 and UL42 each violate the distance constrain of 35 Å and is
therefore colored in light red. All other plotted cross-links meet the DSSO distance
constraint of 35 Å. Predicting virus-virus, virus-host, and host-host dimers of the
structural interactome using AF-Multimer (**b**) with their respective confidence
score distributions (**c**). **d** Per-residue prediction confidence (AF-Multimer pLDDT

score) at the specific inter-linked lysine residue for different types of inter-links. The
lower pLDDT on either lysine residue of the cross-link was used for categorization.
**e** Distribution of inter-link distances (Cα-Cα distance of cross-linked lysines) for AF-
Multimer models in different confidence score ranges. Shown are only inter-links
involving lysines from protein regions with pLDDT scores > 50. Boxes represent
lower and upper quartiles with median marked as horizontal line. Whiskers
represent 1.5 times interquartile range. **f** Examples of well-predicted models
agreeing with the cross-linker distance constraint. No violated cross-links were
observed. See also Supplementary Fig. 4 for prediction of heterodimers using AF3.
Source data are provided.

## System-wide structural insights across HSV-1-host interactions

We initially focused on structural models with inter-links in well-
predicted regions (pLDDT > 50). In these cases, cross-links provide
orthogonal experimental evidence to validate AF-Multimer predic-
tions. Indeed, the average distance bridged by the cross-linker
decreased with increasing model confidence score (Fig. 5e), indicat-
ing that the better the model quality, the better the cross-link distances
fit into the predicted structures. For models with a confidence score
above 0.75 the majority of inter-links were satisfied.

In sum, we obtained 30 models (4 involving viral proteins, 26 host-
host) that have good confidence scores (> 0.75) and agree with at least
half of the inter-links (Supplementary Data 4 and Fig. 5e). Among
those, SHVIP data supported the structural prediction for US12 with
PPIA and UL42, the viral polymerase processivity subunit, binding to its
cellular homolog PCNA (Fig. 5f)[4]. This interaction may explain why
PCNA is enriched at viral DNA replication forks[71].

Furthermore, a substantial number of predicted dimers had high
cross-link satisfaction rates despite lower confidence scores. We
obtained 31 additional models (8 involving viral proteins, 23 host-host)
with confidence scores between 0.5 and 0.75, where the majority of
inter-links from structured regions were satisfied (Supplementary
Data 4). Among these models, we found known HSV-1 PPIs like UL30
with UL42 (confidence score: 0.66), and the ubiquitin-binding protein
US2[72] with ubiquitin (RPS27A) (confidence score: 0.66). This demon-
strates that cross-links can help prioritize realistic models when
AlphaFold quality measures are not sufficient.

To test whether SHVIP is compatible with newer structure mod-
eling tools, we additionally tested its performance using AF3, with
overall similar performance to AF-Multimer (Supplementary Fig. 4a–c).
In general, higher-scoring predictions aligned more closely with the
distance constraints than lower-scoring ones (Supplementary Fig. 4d).
Applying the same model selection criteria across both confidence
categories (see above), we identified two additional models that mat-
ched at least 50% of the detected inter-links from structured regions.
These include STT3A, the catalytic subunit of the ER-resident oligo-
saccharyltransferase complex, known to be crucial for HSV-1
glycosylation[73], with glycoprotein B, and the viral membrane protein
UL45 in complex with PPIB (Supplementary Fig. 4e). These findings
demonstrate that SHVIP is also compatible with AF3, which may out-
perform AF-Multimer for certain targets.

We more specifically investigated the dimer between the tegu-
ment protein UL47 and DDB1 (Fig. 6a), which was one of the best
predicted models with a confidence score of 0.82 in AF-Multimer and
0.77 in AF3. DDB1 functions as an integral component of Cullin 4-RING
ubiquitin ligase (CRL4) complexes[74]. Two UL47-DDB1 cross-links were
identified, with one of them satisfying the distance constraint in the
corresponding AF-Multimer model. In the model, the UL47 C-terminal
region in proximity to the satisfied inter-link adopts an alpha-helical
fold, in positional congruence with previously described viral DDB1
interactors, such as RCMV-E27 (Fig. 6b)[75]. To confirm the UL47-DDB1
interaction and assess the contribution of the UL47 C-terminal helix,
we performed co-immunoprecipitation (IP) experiments. While DDB1
efficiently co-precipitated with UL47-WT, deleting the C-terminal 18

amino acids (676–693) of UL47 was sufficient to disturb this interac-
tion (Fig. 6c). In contrast, the UL47 XL-partner UL48 could still be
immunoprecipitated. Further AP-MS experiments confirmed that the
UL47 C-terminal truncation leads to a strong and specific loss of DDB1-
CUL4A binding (Fig. 6d), indicating that the C-terminal helix acts as a
critical DDB1 interaction module.

Thus, we provide a compendium of virus-host/ virus-virus dimers,
which can be validated by cross-linking distance constraints, reverse
genetics, and orthogonal interaction assays.

## Identifying sequence determinants for virus-host PPIs in IDRs

Most of our inter-links occurred in poorly predicted regions, likely
IDRs. These regions are hotspots for SLiMs that viruses use to interact
with host proteins[6]. AF-Multimer was previously used to identify SLiMs
between proteins in a fragment-based approach outside of infection
contexts[76]. To make the discovery of such interactions sites amenable
for host pathogen, full protein and cross-linking scenarios, we
screened all AF-Multimer models of our cross-linked virus-host PPIs
for short stretches of ordered amino acids (pLDDT > 50) that are in
contact with the respective binding partner and in a disordered
regional context (Fig. 7a). We obtained a list of 29 candidates (Sup-
plementary Data 5) with varying average confidence scores and amino
acid length (Fig. 7b). We manually curated this list to yield 16 sites,
representing a promising set of putative SLiMs mediating virus-host
interactions.

One of the best predicted sites was observed in a short stretch of
amino acids in the N-terminal disordered domain of the alkaline
nuclease UL12 (amino acids 35–50) binding to MAPK8 kinase (also
known as JNK1). Curiously, we observed that 14-3-3 proteins were also
cross-linked to N-terminal lysines within UL12 (K35, K17) (Fig. 7c). In the
MAPK8-UL12 dimer model, the disordered N-terminal domain of UL12
wraps around the MAPK8 kinase domain and binding occurs between a
putative D-motif in the UL12 unstructured region and the MAPK8
docking groove (Fig. 7d)[77]. We therefore mutated the candidate
D-motif (P39A, L44A, Fig. 7e). Comparing mutant to wildtype UL12
interactomes indicates that the mutations result in loss of MAPK8
binding but stronger association with other UL12 interactors (14-3-3
proteins).

We hypothesized that both MAPK8 and 14-3-3 may bind to the
same short sequence stretch within UL12. Using a sequence-based
prediction tool for 14-3-3 binding motifs[78] we identified Serine 41 as
best hit for 14-3-3 interaction (Supplementary Fig. 5a). Following this,
we mutated critical amino acids in the viral genome (S41A, P43A) to
disrupt 14-3-3 binding. We then compared wildtype to mutant in an AP-
MS experiment using SILAC (stable isotope labeling of amino acids in
cell culture) labeling (Supplementary Fig. 5b and Fig. 7f). Wildtype
UL12 co-precipitated with 14-3-3 proteins stronger than S41A, P43A-
UL12. In contrast, the mutant interactome showed enrichment of
several heat-shock proteins and histones, suggesting effects on UL12
folding or stability upon loss of 14-3-3 binding. Taken together, our
analysis identifies two SLiMs that co-exist on a narrow stretch of amino
acid within the UL12 N-terminal disordered domain and enable UL12 to
recruit MAPK8 and 14-3-3 proteins.

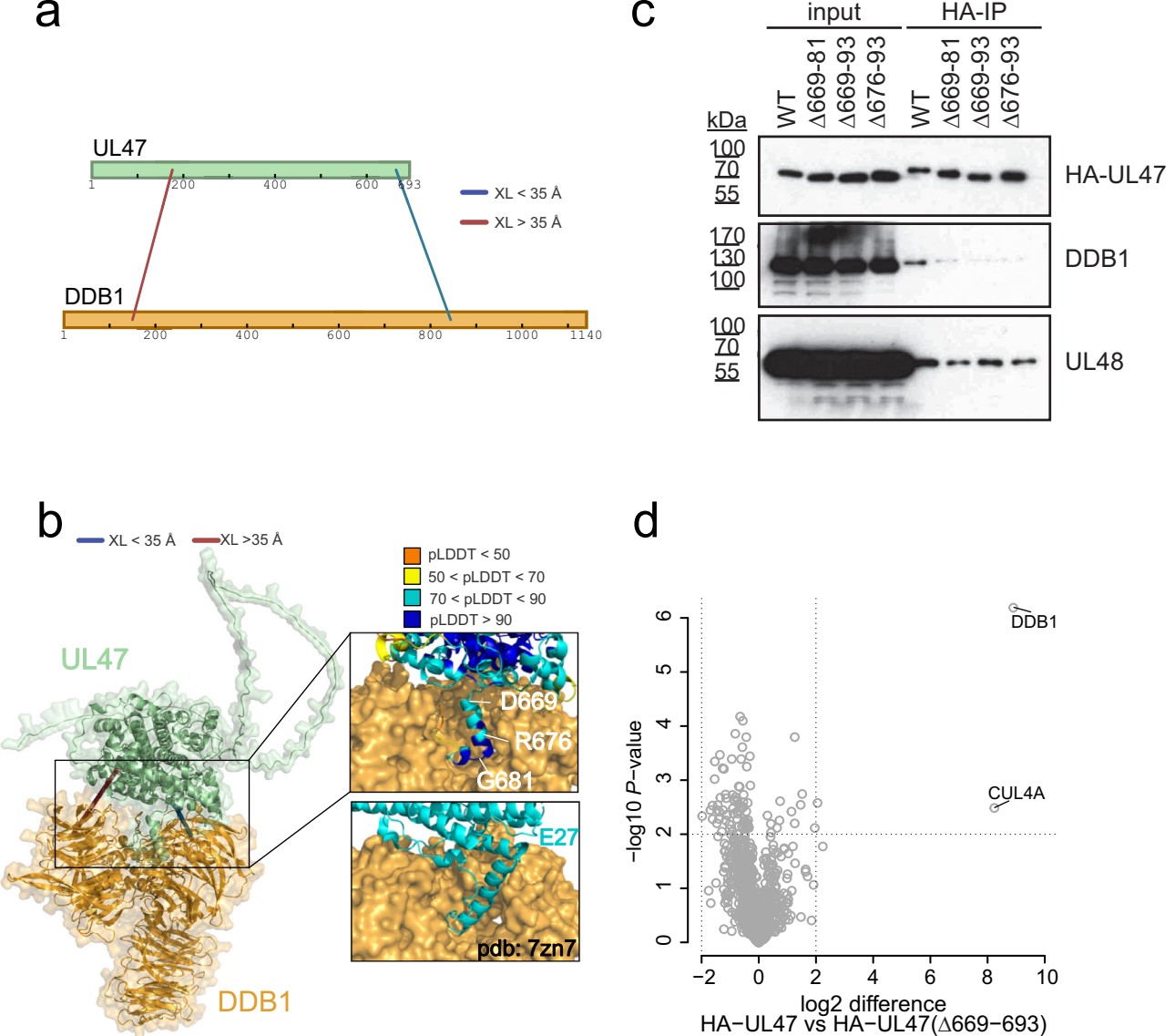

**Fig. 6 | Structural determinants of the DDB1-UL47 heterodimer. a** UL47-DDB1 cross-links mapped onto sequence bars[109]. **b** DDB1-UL47 cross-links mapped onto the predicted dimer with color coding of the cross-links according to distances between Cα atoms. The inset shows the AF-Multimer prediction confidence for the C-terminal helix of UL47. Displayed below the inset is the experimentally resolved structure of E27-DDB1 (pdb 7zn7) for comparison. **c** HA-directed Co-IPs against different mutant and wildtype variants of UL47 from infected cells at 24 hpi (representative result of *n* = 2 experiments). **d** AP-MS experiments directly comparing the interactome of wildtype UL47 to mutant UL47 in a label-free set-up. *P* values are based on two-sided *t* tests without multiple hypothesis correction based on *n* = 3 biological replicates. Source data are provided.

In addition, our list of putative SLiMs includes other compelling predictions. For example, we found the interaction of gE with the adaptor protein AP1M1, with a putative tyrosine-trafficking motif in gE (Supplementary Fig. 6a). Also, we predicted the central player in autophagy MAP1LC3B (also known as LC3-B) interacting with UL49, which has a putative LC3 interaction region (LIR)[79] (Supplementary Fig. 6b). Further, we observed the interaction between UL49 and the cellular Phosphatase PP-1 (also known as PPP1CA), for which the predicted model indicated binding of an amino acid stretch in the unstructured region of UL49 to the RVXF-motif recruitment site in PP-1 (Supplementary Fig. 6c). As the corresponding amino acid sequence in UL49 suggests the presence of a non-canonical RVXF motif (Supplementary Fig. 6d), we aimed for independent validation of this binding site. Indeed, we found that mutating the motif in UL49 (R52A, F57A) reduces PP-1-UL49 binding in HA-directed AP-MS assays (Supplementary Fig. 6e, f). These findings illustrate that combining SHVIP and AF-

Multimer-based structure predictions allows the discovery of critical sequence determinants of virus-host PPIs within IDRs.

## Discussion

While interaction proteomics has contributed immensely to uncover the molecular processes underlying viral pathogenesis[23–33,80–83], a global view of interaction contact sites within infected cells has remained elusive. The SHVIP approach introduced here addresses this shortcoming. Using SHVIP, we define the structural interactome of a complex herpesvirus in intact infected cells, revealing distinct clusters of spatially separated virus-host PPIs. Orthogonal validations, structure predictions, and molecular genetics provide insights into the structural basis of a subset of these interactions.

XL-MS integrates well with AF to inform structural predictions at interactome scale[84–86] delivering residue-residue connections as corroborative evidence to purely in silico analyses. Enhancing

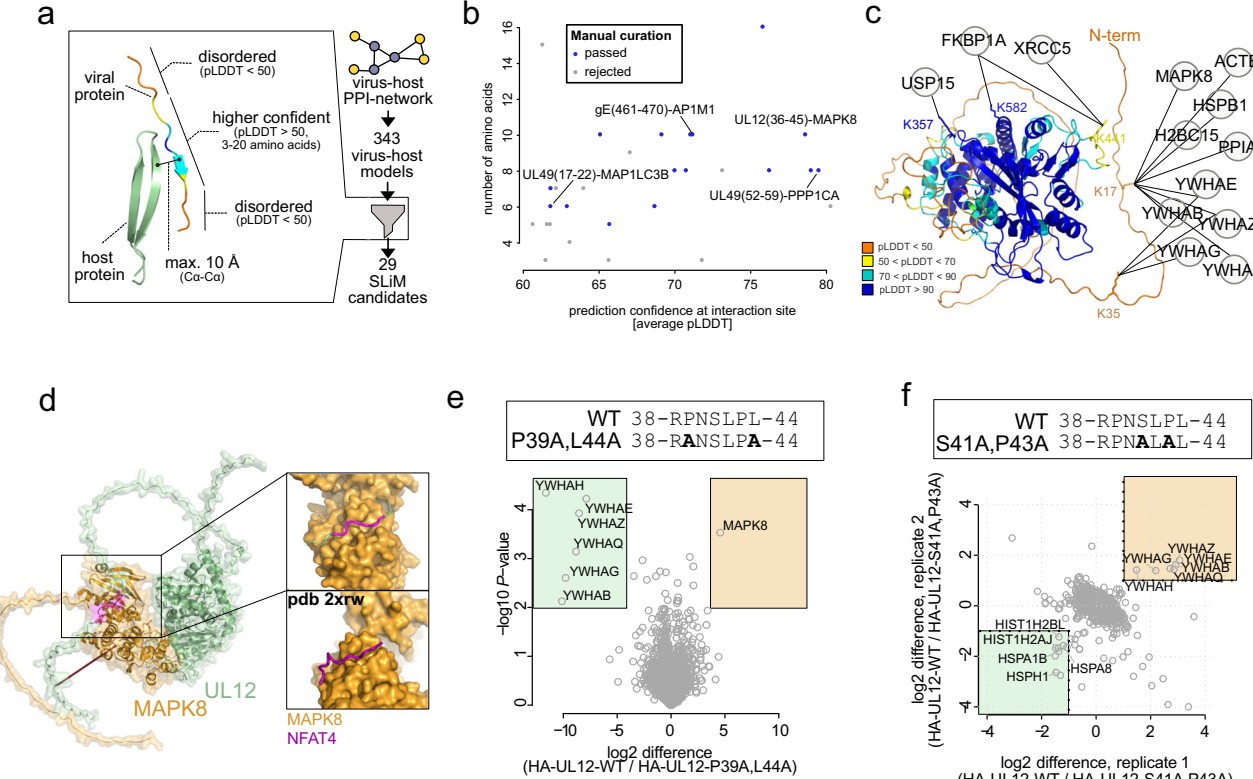

**Fig. 7 | Identifying sequence determinants for host-virus interactions in IDRs.**
**a** Workflow for selecting putative short linear interaction sites in IDRs from AF-Multimer models. **b** Comparison of the residue length for the putative interaction motif to the confidence of the predicted interface (average pLDDT of all amino acids within the selected site). **c** Host proteins cross-linked to UL12, depicted on the monomeric AF2 model of UL12, color coded by pLDDT. **d** Predicted MAPK8-UL12 dimer with a putative D-motif highlighted magenta. The inset shows the magnified region of the D-motif in UL12 interacting with the D-site in MAPK8. Shown below the inset is a solved crystal structure of MAPK8 with the D-motif containing peptide of NFAT4 (pdb 2xrw). **e** Mutational disruption of the D-motif in UL12 in the viral genome of HSV-1, creating P39A, L44A-UL12. Label-free comparative AP-MS of the

wildtype UL12 and P39A, L44A-U12 interactomes. The p-values are based on two-sided *t* tests without multiple hypothesis correction of *n* = 3 replicates. Boxed regions highlight proteins that interact significantly stronger with P39A, L44A-UL12 (shaded green) or WT-UL12 (shaded orange), based on a cut-off at *P* = 0.01 and log2 fold-change of 4. **f** Mutational disruption of the 14-3-3 binding site in UL12 in the viral genome of HSV-1, creating S41A, P43A-UL12. SILAC-based comparative AP-MS of the wildtype UL12 to the S41A, P43A mutated UL12 interactome, with both (*n* = 2, label-swap) replicates depicted. Boxed regions highlight proteins that interact stronger with S41A, P43A-UL12 (shaded green) or WT-UL12 (shaded orange), based on a log2 fold-change cut-off of one in both replicates. Source data are provided.

interactome-wide AF modeling with SHVIP is particularly valuable from a virological point of view because (1) AF-only predictions of dimers involving viral proteins have generally lower quality than those of host proteins[70] (Figs. 5 and 2). Experimental structures of host-virus protein complexes are scarce for HSV-1 and many other viruses. Augmenting AF with XL-MS data proved to be effective to cross-validate generally lower-scoring host-virus protein complex predictions and thus identify convincing structures, which are, in this case, potentially linked to DNA replication, immune evasion, and signaling. For instance, we found a C-terminal helix in the tegument protein UL47 that interacts with the CRL4 adapter unit DDB1 (Fig. 6) in a manner that is typical for DCAF-type substrate receptors of CRL4. It will be interesting to see whether UL47 acts in analogy to other viral DCAF mimics[75] and exploits CRL4 to target antiviral host proteins for proteasomal degradation. To arrive at even more confident predictions, it will be promising to improve structure modeling, as e.g., through approaches that incorporate cross-linking distance constraints[87] or massive sampling strategies[88,89] in the future.

SHVIP can capture weak and context-sensitive PPIs because it capitalizes on protein cross-linking within intact cells. Such interactions may be disrupted during cell lyses and therefore elusive to AP-MS. While SHVIP allows characterizing viral proteins that are challenging to study outside their cellular context (such as proteins undergoing LLPS[90] or transmembrane proteins[91]), the sensitivity of XL-MS

workflows depends a lot on the abundance and structural complexity of the specific protein. Although our approach could enrich viral proteins, it does not eliminate identification bias originated from protein abundances. When interpreting XL-MS data, it is therefore critical to keep in mind that interactions for abundant proteins are more likely to be covered compared to low-abundant proteins. This abundance bias is reduced in AP-MS, which is because detecting linear peptides and focusing on interactors of one individual protein may increase sensitivity. Beyond sensitivity considerations, AP-MS is primarily based on protein affinity, whereas XL-MS is driven by protein proximity. These methodological differences likely explain why both techniques provide overlapping information while also offering a substantial degree of complementarity (Fig. 4). XL-MS captures a snapshot of the in situ occurring interactome that preserves the context of higher-tier interactors. It is therefore better able to capture PPIs only existing in specific subcellular contexts, such as LLPS-based condensates. For instance, we identified UL49 as the most connected viral protein in the XL-MS-based interaction network, which is in agreement with previous findings that UL49 interacts with a variety of viral and cellular proteins[61,63,64,92], such as histones[66–68]. It is possible that UL49 achieves the organization of many PPIs through its ability to undergo LLPS[20].

We observed that cross-links involving viral proteins are enriched in IDRs (Fig. 5d), recognized hotspots for SLiMs frequently exploited

by viruses to manipulate host processes[6]. Based on XL-supported heterodimers and their corresponding AF-Multimer models, we propose 16 potential SLiMs within IDRs (Fig. 6b). We validated the non-canonical RVXF-motif in the UL49 protein as a SLiM for binding PP-1 phosphatase. In analogy to HCMV-UL32[93], UL49-PP-1 interaction may be important to regulate the tegument phosphorylation state[94], facilitating virion assembly. Furthermore, we identified two co-existing SLiMs within the disordered segment of alkaline nuclease UL12 responsible for competitive binding of MAPK8 and 14-3-3 proteins. These interactions might constitute a regulatory switch, allowing HSV-1 to link the viral replication machinery to MAPK8, a potent effector of cellular stress signaling that is crucial for reactivating HSV-1 from latency[95]. These examples show that canonical and non-canonical interaction motifs are discoverable by integrating SHVIP and AF-Multimer. The distinctive advantage of this approach lies in its incorporation of structural context and co-evolutionary relationships, enabling a significant improvement over sequence-based methods for predicting SLiMs[7].

The methodology is, in principle, applicable to study native cells infected with any virus inducing host shutoff[41]. This includes medically or veterinary relevant viruses such as influenza A viruses[42,96], poxviruses[97], picornaviruses[98], flaviviruses[99], or asfarviruses[45]. Also, it does not require upfront genetic work, which makes it readily applicable to emerging viruses and viruses that do not tolerate epitope tagging in their genome. However, it is important to consider that HPG-labeling requires methionine starvation, impacting cell viability[100] and viral titers[101], particularly when employed for prolonged periods. This may, in the future, be mitigated by adopting starvation-free protocols[100] for SHVIP.

In summary, SHVIP offers various conceptual advantages for virus-host PPI profiling. Our data on a complex and medically relevant herpesvirus deliver a blueprint for the structural characterization of PPIs within their intact environment during viral infections. The method unlocks an opportunity for the in-depth characterization of viral gene functions during cellular infections.

## Methods

### Cells and viruses

Human embryonic lung fibroblasts (HELFs) were maintained according to established protocols[102], using Eagle's minimum essential medium (EMEM) supplemented with Earle's balanced salt solution, 25 mM HEPES, 1 mM sodium pyruvate, 2 mM L-alanyl-L-glutamine, nonessential amino acids, 0.75 ‰ (w/v) sodium bicarbonate, 50 μg/ml gentamicin and 10 % fetal bovine serum (FBS) and used for preparation of viral stocks. Recombinant HSV-1 strain 17[103] was used for all experiments. Infectious virus titerss were determined by immunotitration and indicated as immediate early (IE)-forming units (IU) per mL. In brief, HELFs were infected with serial dilutions of cell-free virus stocks. At 8 hpi, the number of infected cells was determined by flow cytometry of IE antigen ICP4 expression, using a 1:200 dilution of the mouse anti-ICP4 antibody clone H943 from Santa Cruz Biotechnology (sc-69809) and an appropriate fluorescently labeled anti-mouse IgG antibody. Viral mutants were created by traceless bacterial artificial chromosome mutagenesis according to established protocols[104]. Mutations were verified by Sanger sequencing and PCR. See Supplementary Data 6 for a list of mutagenesis primers. For all infection experiments, a multiplicity of infection of 5 IU/cell was used.

### Stable isotope labeling in cell culture

HELFs were cultured for at least five passages in SILAC-DMEM medium supplemented with 10 % (v/v) dialyzed fetal bovine serum (FBS, Pan-Biotech), 1 × Glutamax (Gibco), 1 × non-essential amino acids (Gibco), 100 Units/mL Penicillin (Gibco), and 100 μg/mL Streptomycin (Gibco). For heavy labeling medium was supplemented with 0.8 mmol/L L-

[13C6,15N2]-Lysine (Lys8) and 0.4 mmol/L L-[13C6,15N4]-Arginine (Arg10). For light labeling medium was supplemented with natural Lysine and Arginine to the same concentrations, respectively.

### Co-immunoprecipitation

Cells were harvested at 24 h post infection and extracted by freezing-thawing in immunoprecipitation buffer, containing 50 mM Tris–Cl pH 7.4, 150 mM NaCl, 10 mM MgCl$_2$, 1 mM NaF, 10 mM β-glycerophosphate, 0.5 mM Na$_3$VO$_4$, 10 mM N-ethylmaleimide, 0.5% Nonidet P-40 (NP-40), 10% glycerol, 1 mM dithiothreitol (DTT), 0.3 μM aprotinin, 23 μM leupeptin, 1 μM pepstatin, 0.1 mM Pefabloc. HA-UL47-containing protein complexes were immunoprecipitated by incubating extracts with HA-specific antibodies and protein G-coupled Sepharose beads (4 Fast Flow, GE Healthcare). The sepharose-bound proteins were analyzed by standard immunoblotting for the presence of HA-UL47, DDB1, and UL48. The following antibodies were used: anti-DDB1 (rabbit polyclonal, Bethyl Laboratories, A300-462A), anti-HA (rat clone 3F10, Roche, ROAHAHA), anti-UL48 (mouse clone 1–21, Santa Cruz Biotechnology, sc-7545).

### Affinity-purification mass spectrometry

Eight viral baits were selected from our network for subsequent orthogonal validation by AP-MS. We excluded viral glycoproteins, capsid or capsid-associated proteins, and proteins with fewer than two PPIs. From the remaining shortlist, we selected three viral proteins with few PPIs (<10 PPIs: UL48, DBP, DUT), three with moderate number of PPIs (>10 and <50 PPIs: UL12, NEC2, UL47), and two highly inter-linked proteins (UL49, US11). AP-MS was carried out according to established protocols[27]. Label-free experiments were performed in triplicates, and SILAC-based experiments were performed in label-swap duplicates. In all cases, cells were harvested 24 h after infection with wild-type or mutant virus. One fully confluent 15 cm dish of HELFs per replicate and condition was used. Following washing with PBS, cells were lysed by a 30 min incubation in lysis buffer: 25 mM Tris-HCl (pH 7.4), 125 mM NaCl, 1 mM MgCl$_2$, 1% NP-40, 0.1% sodium dodecyl sulfate (SDS), 5% glycerol, 1 mM DTT, 0.3 μM aprotinin, 23 μM leupeptin, 1 μM pepstatin, 0.1 mM Pefabloc. Lysates were cleared by centrifugation, and the cleared lysates were incubated with anti-HA magnetic microbeads (Miltenyi) for 60 min before being applied to μMACS microcolumns (Miltenyi). We used lysis buffer for the first washing step, lysis buffer without detergent for the second, and 25 mM Tris-HCl (pH 7.4) for the final washing step. Protein material was eluted in a total volume of 0.2 mL 8 M guanidine hydrochloride at 95 °C and precipitated from the eluates by adding 1.8 mL LiChrosolv ethanol and 1 μL GlycoBlue. After incubation at 4 °C overnight, samples were centrifuged for 1 h at 4 °C, and ethanol was decanted before samples were subjected to sample preparation.

### SHVIP sample preparation

Cells were infected at a multiplicity of infection of 5 IU/cell, using standard protocols. At 7 h post infection, cells were washed twice with pre-warmed PBS before the L-HPG-labeling medium was added, consisting of 500 μM HPG (Click Chemistry Tools), methionine-free DMEM (Gibco ref.), and 10% dialyzed fetal bovine serum (Pan Biotech). Cells were washed in PBS and harvested by scraping at 24 hpi before being cross-linked with 5 mM DSSO in a 1:1 (vol/vol) PBS/cell suspension for 1 h at room temperature under constant shaking. Afterwards, cells were lysed in a buffer containing 200 mM Tris (pH 8.0), 4% 3-((3-cholamidopropyl) dimethylammonio)-1-propanesulfonate, 1 M NaCl, 8 M urea, and protease inhibitors. Cell lysates were incubated for 30 min at 4 °C. Genomic DNA in the samples was digested by addition of Benzonase. Afterwards, cell lysates were sonicated in a Bioruptor Pico (Diagenode) with 10 cycles of 30 s sonication pulses, followed by 30 s without sonication. Samples were centrifuged at 10,000 × g for 10 min, and the resulting pellet was discarded. Approximately 5% of the

sample volume was saved as input, while the remainder was subjected to enrichment of labeled proteins.

HPG-labeled proteins were enriched using copper(I)-catalyzed azide-alkyne cycloaddition, also known as click reaction, using the Click-&-Go protein enrichment kit (Click Chemistry Tools) according to the manufacturer's protocol. In brief, samples were incubated overnight rotating together with picolyl-azide conjugated agarose beads and a copper(I)-ion containing catalyst solution. Protein-conjugated agarose beads were centrifuged at $1000 \times g$ for 1 min, and the supernatants containing unbound, unlabeled cellular proteins were discarded. Afterwards, disulfide protein bridges were reduced by adding 10 mM DTT and alkylated by adding 40 mM chloroacetamide (CAA). Subsequently, agarose beads were transferred to 0.8 mL Pierce centrifuge columns (Thermo Scientific). Beads were subjected to a five-step washing protocol employing 10 times 500 µl of the following washing buffers: (i) 1 % w/v SDS, 250 mM NaCl, 5 mM EDTA in 100 mM Tris (pH 8.0); (ii) 8 M urea in 100 mM Tris (pH 8.0); (iii) 80 % v/v acetonitrile (ACN) in water; (iv) 5 % v/v ACN in 50 mM triethylammonium bicarbonate (TEAB); (v) 2 M urea plus 5 % v/v ACN in 50 mM TEAB. After the last washing step, beads were resuspended in the final washing buffer. To release the captured proteins from the beads, proteins were digested by addition of trypsin and lysyl endopeptidase C (Lys-C). Samples were incubated at 37 °C overnight shaking. The peptide-containing supernatants were recovered by 5 min centrifugation at $500 \times g$, and proteolytic digestion was stopped by addition of 1% formic acid (FA). Peptides were desalted using Sep-Pak C8 1cc vacuum cartridges (Waters) according to manufacturer's protocol. Proteins from input samples were extracted using methanol-chloroform precipitation. Precipitated proteins were resuspended in a buffer containing 1% w/v SDC, 5 mM Tris-(2-carboxyethyl)phosphin (TCEP), 40 mM CAA in 50 mM TEAB (pH 8.0). The following digestion by trypsin and Lys-C was conducted as described above.

### Off-line fractionation

Desalted peptides were further fractionated by strong cation exchange (SCX) chromatography using an Agilent 1260 Infinity II HPLC system equipped with a PolySULFOETHYL A column (PolyLC). Peptides were separated using a 95 min gradient ranging from 100% Buffer A (20% ACN in water) to a mixture of 20% Buffer A and 80% Buffer B (5 M NaCl in 20% ACN and water), collecting 45 s fractions. Fractions were desalted using C8 stage tips[105] and dried by speed vacuum. Dried samples were stored at −20 °C before LC-MS measurement.

### AlphaFold predictions

Hetero-dimers were predicted by AF-Multimer v2.3[8] using the default protocol and database sequence search methods. pTM, ipTM, confidence scores (0.8*ipTM+0.2*pTM), and predicted alignment errors were extracted from.pkl files. Of each dimeric prediction, we selected the model with the highest AlphaFold confidence score and highest cross-link agreement for further analysis. Predicted alignment errors were plotted using Python. Cross-links were plotted on predicted structures using the bio3D package in R Studio. Models with at least half of cross-links in-range (<35 Å), taking into account only links from well-structured regions (pLDDT > 50), were included in Supplementary Data 4. The AF2 model of HSV-1-UL12 was downloaded from https://www.bosse-lab.org/herpesfolds/.

Hetero-dimeric PPIs were additionally predicted by AlphaFold3 v.3.0.1[9] on a local workstation utilizing an NVIDIA RTX A6000. Structure prediction was done in a Docker environment with default parameters for protein-protein sequences, generating 5 predicted models through 1 random seed; model parameters were provided by Google Deepmind.

To identify interaction sites within disordered regions, we developed a two-step algorithm in Python 3.10 utilizing Biopython[106]. For each predicted dimeric structure (all ranks), residues of a chain are extracted if their corresponding C-alpha coordinates exhibit a Euclidean distance of less than 10 Å to C-alpha coordinates of the other chain. Following this, we investigated these stretches of residues for their pLDDT scores. As a pLDDT below 50 is a predictor of disorder[107], consecutive residues with a motif length of three to up to 20 amino acids are extracted if each of them exhibits a pLDDT above 50 and if they are adjacent to residues with a pLDDT below 50. We then excluded motifs with an average pLDDT below 60 to retain higher confidence predictions.

Structural images were generated using PyMOL (Schrödinger, LLC, version 2.4).

### LC-MS of XL-MS data

Cross-linked peptides were measured on an Orbitrap Fusion Lumos Tribrid system (Thermo Fisher Scientific) equipped with a FAIMS Pro Duo interface (Thermo Fisher Scientific) operating with Xcalibur 4.6 and Tune 4.0. For this, ~1 µg of sample was loaded with an online-connected Ultimate 3000 RSLC nano LC system (Thermo Fisher Scientific) onto a 50 cm analytical, in-house packed reverse-phase column (Poroshell 120 EC-C18, 2.7 µm, Agilent Technologies) and separated with a 180 min or 120 min gradient going from 0.1% w/v FA in water (buffer A) to 0.1% w/v FA in 80% v/v ACN (buffer B) at a flow rate of 250 nl/min. FAIMS compensation voltages were alternated between −50, −60 and −75 V.

We conducted two experiments with different MS acquisition schemes (MS2-MS3 and MS2-only) to provide evidence of the performance of the workflow when using different acquisition schemes. For the SHVIP experiment with MS2-only acquisition scheme, cross-linked peptides were detected using a stepped-higher energy collision-induced dissociation (HCD)-MS2 method, where for MS1, Orbitrap resolution was set to 120,000 with a scan range of 375–1600 $m/z$ and mass range set to Normal. Standard parameters were used for automatic gain control (AGC), and 50 ms of injection time was used. Precursors of charge states 4-8 were selected with an isolation window of 1.6 $m/z$ and fragmented by stepped-HCD with the energies 21%, 27% and 33%. Dynamic exclusion was enabled with an exclusion period of 60 s. Cycle time in between master scans was set to 2 s. For MS2, Orbitrap resolution was set to 60,000 with mass range set to Normal and scan range to Auto. Maximum of injection time was adjusted to 118 ms, and AGC target to 200%.

For the SHVIP experiment with the MS2-MS3 acquisition scheme, cross-linked peptides were detected by a stepped HCD-MS2-collision-induced dissociation (CID)-MS3 method. MS1 and MS2 scan parameters were the same as for the stepped-HCD-MS2-only method. MS3 scans were triggered when DSSO signature peaks with a mass difference of 31.9721 mass units were detected. The two most intense reporter peaks with charge states 2–6 were selected as precursors for MS3 with an isolation window of 2 $m/z$ and fragmented by CID with a collision energy of 35% and an activation time of 10 ms. MS3 scans were acquired in the Ion Trap with scan rate set to Rapid, 100 ms injection time, AGC target set to 200%, mass range set to Normal, and scan range set to Auto.

### LC-MS of bottom-up data

Bottom-up proteomic samples were measured on an Orbitrap Fusion Tribrid instrument (Fusion), an Orbitrap Fusion Lumos instrument (Lumos), an Orbitrap Elite instrument (Elite) (all Thermo Fisher Scientific) connected online to an Ultimate 3000 RSLC nano LC system (Thermo Fisher Scientific) or an Orbitrap Exploris 480 (Exploris) connected online to a Vanquish neo UHPLC system (Thermo Fisher Scientific). Peptides were separated on an in-house packed 50 cm analytical, reverse-phase column (Poroshell 120 EC-C18, 2.7 µm, Agilent Technologies) with 120 min or 180 min gradients at a flow rate of 250 nl/min.

Methods run on Fusion or Lumos were measured with a HCD-MS2 method with MS1 scans acquired in the Orbitrap with the resolution set to 120,000, while MS2 spectra were acquired in the Ion Trap. Cycle time in between master scans was set to 1 s, and dynamic exclusion was set to 40 s. Intensity threshold was set to 1E04 for Fusion and to 5E03 for Lumos, and maximum injection time to 50 ms, while AGC target was kept at standard settings. Precursors of charge state 2–4 were selected for fragmentation with a precursor isolation window of 1.6 $m/z$ and fragmented with the HCD energy of 30 %.

For measurement on Exploris, peptides were measured with an HCD-MS2 method with MS1 scans acquired in the Orbitrap with the resolution set to 120,000. Cycle time in between master scans was set to 2 s, and dynamic exclusion was set to 40 s. Intensity threshold was set to 1E04, and maximum injection time was set to Auto. AGC target was set to 300%. Precursors of charge state 2–4 were selected for fragmentation with a precursor isolation window of 1.6 $m/z$ and fragmented with the HCD energy of 30%. MS2 spectra were acquired in the Orbitrap with a resolution set to 15,000, with AGC target set to Standard and maximum injection time set to Auto.

Measurements run on Elite were acquired using a CID-MS2 method. MS1 scans were acquired in the Orbitrap with resolution set to 120,000 and mass range from 350 to 1500 $m/z$. Dynamic exclusion was enabled with a duration of 60 s. Top 15 precursors of charge states 2 and 3 were selected with an isolation window of 1 $m/z$ for fragmentation by CID. Normalized collision energy of 35%, activation time of 10 ms, and activation Q of 0.25 were used for CID fragmentation. MS2 spectra were acquired in the Ion trap with mass range set to Normal and scan rate set to Rapid.

## Cross-link data analysis

Peak lists (.mgf files) were generated in Proteome Discoverer (v.2.1) to convert .raw files into .mgf files containing HCD-MS2 or HCD-MS2-MS3 data, respectively. The .mgf files were searched using a stand-alone search engine based on XlinkX v.2.0[108] with the following settings: MS ion mass tolerance, 10 ppm; MS2 ion mass tolerance, 20 ppm; fixed modification, Cys carbamidomethylation; variable modification, Met oxidation; enzymatic digestion, trypsin; allowed number of missed cleavages, 3; DSSO cross-linker, 158.0038 Da (short arm, 54.0106 Da; long arm, 85.9824 Da). Reaction sites at lysine residues and protein N-termini were considered. Spectra were searched against a concatenated target-decoy database generated on the basis of the virus and host proteome determined by bottom-up proteomics. Raw files from both experiments (MS2-only and MS2-MS3 acquisition) were searched separately, results were then combined, and the FDR at 1% was applied at the level of residue-residue connections, separately for intra- and inter-links on the basis of a target-decoy competition strategy using randomized decoys. Following this, all cross-links were retained when they matched to a viral protein or to host protein with a viral interaction partner (virus-centric network). This network still contains host-host cross-links between host proteins that are also linked to viral proteins. The PPI-level FDR in this set of PPIs was assessed based on the ratio of decoys to targets in the virus-centric network. The resulting network was clustered with the edge-betweenness algorithm[47] using in-house R scripts and the edge.betweenness.community function from igraph package. Clusters and networks were visualized in Cytoscape v3.7.2 or using xiNET[109]. The comparison of PPI and cross-link identifications between input and enriched samples is based on individual searches as described above, but with a naive 1 % FDR (that is, FDR filtering applied on the combined set of intra- and inter-links).

## Bottom-up proteomics data analysis

AP-MS experiments were designed as triplicate experiments comparing anti-HA APs in lysates containing the transgene to control purifications of the wildtype strain, or in case of mutation of short linear interaction motifs, comparing HA-tagged wildtype protein containing virus against the HA-tagged mutant protein containing virus. The set-up was label-free, and control experiments were performed and measured in parallel. Raw files were analyzed using MaxQuant v.2.0.3.0[110], with standard parameters and match between runs, iBAQ, and LFQ enabled. The HSV-1 reference proteome UP00009294 and Uniprot reference proteome of human protein sequences (downloaded 2020) was used. FDR cut-offs were set at 1% at PSM, protein, and modification site levels. The proteinGroups.txt file was used for subsequent analysis with potential contaminants, reverse database hits, and proteins only identified by a modification site removed. LFQ intensities were log2 transformed and, when appropriate, missing values were imputed on the basis of a normal distribution shrunk by a factor of 0.3 and downshifted by 1.8 standard deviations only when a protein was quantified in all three replicates of either experiment or control. Log2 fold-changes and p-values from a $t$ test were calculated based on these values. The significance of overlap between AP-MS and XL-MS was calculated using phyper function in R.

SILAC-quantified AP-MS experiments were conducted in duplicates, comparing HA-tagged wild-type target protein containing virus with HA-tagged mutant target protein containing virus. Raw files were analyzed using MaxQuant v. 1.6.2.6. Quantification was based on MaxQuant normalized SILAC ratios using two parameter groups in the analysis for light and heavy (Lys8 and Arg10) labeled proteins, respectively. Remaining parameters were kept as described above, and the re-quantify option was enabled. The proteinGroups.txt file was used for data evaluation in R with log2-transformed SILAC ratios. Proteins enriched or depleted over a log2 ratio of 1 in both replicates were considered as hits.

## Reporting summary

Further information on research design is available in the Nature Portfolio Reporting Summary linked to this article.

## Data availability

The mass spectrometry proteomics data have been deposited to the ProteomeXchange Consortium via the PRIDE[111] partner repository with the dataset identifier PXD047422. AlphaFold models are available via figshare under: https://doi.org/10.6084/m9.figshare.24639279.v2 (AF2.3) or https://doi.org/10.6084/m9.figshare.29064041.v1 (AF3). A summary on the performed experiments and raw files is available in Supplementary Data 7. Source data are provided with this paper.

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

## Acknowledgements

The authors thank Beate Sodeik (MHH, Hannover) for providing us with recombinant HSV-1 strain 17 and Philip Lössl (Absea Biotechnology, Berlin) for critically reviewing and editing the manuscript. Funding was provided by the Deutsche Forschungsgemeinschaft (DFG) grant BO 5917/1-1 (B.B.), Leibniz-Wettbewerb (K284/2019) (F.L.). Financial support to A.E. was obtained from the Swedish Research Council for Natural Science, grant No. VR-2016-06301 and Swedish E-science Research Centre, and from Knut and Alice Wallenberg Foundation. Computational resources to A.E. were obtained by the Swedish National Infrastructure for Computing via grants: SNIC 2021/5-297, SNIC 2021/6-197, Berzelius-2021-29, and Berzelius-2022-106.

## Author contributions

B.B. conceptualized the project. B.B., L.M., I.G., J.R., L.W., and F.L. developed the methodology. B.B., L.M., I.G., B.V., J.R., A.E., and L.W. conducted the investigations. B.B., L.M., I.G., and L.W. conducted formal analysis. I.G., B.V., L.W., A.E., and F.L. procured resources. B.B. and L.M. performed visualization. B.B. and L.M. curated the data. B.B., L.W., and F.L. acquired funding. B.B., L.W., and F.L. administered the project. B.B., L.W., and F.L. supervised the project. B.B. and L.M. wrote the original draft. B.B., L.W., and F.L. reviewed and edited the manuscript.

## Funding

## Competing interests

F.L. is a scientific advisor to Absea Biotechnology and vantAI. The remaining authors declare no competing interests.
