## [Transparent Peer Review file · Nature Communications]

Structural Host-Virus Interactome Profiling of Intact Infected Cells

Corresponding Author: Professor Fan Liu

Version 0:

Reviewer comments:

Reviewer #1

(Remarks to the Author)

Bogdanow et al. investigate the interaction landscape of the HSV-1 infection cycle, effectively combining a specialized labeling and cross-linking strategy with structural modeling to propose viable human-virus complexes. The study presents a valuable set of interactions that could be leveraged by the scientific community to develop therapeutic components aimed at disrupting the HSV-1 infection cycle. However, before publication, the manuscript should be revised in light of the following computational improvements:

- As the authors mention, human-viral and viral-viral protein complexes lack the typical coevolution signals utilized by AlphaFold. In such cases, extensive sampling has been shown to mitigate the limitations caused by shallow MSA profiles. Therefore, the authors should explain why they adhered to a single version of AlphaFold2 without experimenting with sampling parameters/weights to potentially improve the quality of their models. Furthermore, they should clarify why they did not use AlphaFold3 for these modeling tasks, which could offer an opportunity to enhance model quality over the current approach.
- While pDockQ is a useful metric for predicting complex quality, its familiarity is largely limited to the computational biology community. To improve the accessibility and clarity of their results, the authors are encouraged to also report confidence scores, as well ipTM or iPAE, which are more widely recognized and understood across the broader scientific community.
- Additionally, the authors are advised to provide an analysis of Neff (the effective number of sequences in the MSA) versus the predicted quality of the models. This would offer insights into how the depth of the MSA impacts the quality of the generated models, providing a clearer understanding of the relationship between sequence diversity and model accuracy.

Reviewer #2

(Remarks to the Author)

In this paper, the authors present an innovative approach for comprehensively detecting virus-host protein-protein interactions (PPIs) without cell lysis in the affinity purification of protein complexes, which is a conflicting procedure to detect PPIs. The combination of crosslinking mass spectrometry and selective bio-orthogonal labeling of viral proteins represents an elegant and sophisticated strategy. Using this newly developed method, the authors identified more than 700 PPIs in HSV-1-infected cells. Furthermore, their analyses of cross-linked peptide data provided domain-specific insights into PPIs, potentially improving the accuracy of predicting interactions between structured and disordered regions. These data will contribute significantly to advancing our understanding of HSV-1 infection. The concept of this study is intriguing and can be easily applied to other viruses. The authors present cutting-edge data supported by advanced technology. On the other hand, there seem to be concerns regarding the comprehensiveness and specificity of the data presented. In addition, this study did not thoroughly evaluate the biological significance of the newly identified HSV-1-host interactions in the context of viral infection. Such data will greatly strengthen the results of this study and encourage researchers in the broader virology research community to utilize this newly developed approach.

Major Comments:

1. The authors should further investigate the biological significance (that is, contribution to viral replication and/or pathogenesis) of at least one of the newly identified HSV-1-host interactions in the context of viral infection.

2. The authors stated, "Together, this indicates that SHVIP sensitively captures a large part of the available viral proteome." Given that HSV-1 expresses approximately 100 viral proteins in infected cells, detecting only 46 HSV-1 proteins in this dataset does not convincingly demonstrate the coverage of a substantial portion of the viral proteome. The interactions between HSV-1 and host proteins are likely far more complex, and this study may have uncovered only a small fraction of these interactions.

3. The authors must explain more clearly how they ensured the specificity of their data. Although crosslinkers are designed to react with specific amino acids, nonspecific binding can still occur, potentially resulting in false positives. It is crucial to specify the extent to which false positives were accounted for, and how they were addressed or excluded from the analysis.

4. Crosslinking between peptides generates irregular fragmentation patterns, making the mass spectra of cross-linked peptides significantly more challenging to analyze than those of regular peptides. Furthermore, the presence of various post-translational modifications in HSV-1-infected cells complicates the identification of crosslinked peptides and determination of crosslinking sites. Given these complexities, it is reasonable to question whether a standard FDR threshold of <1%, typically used in conventional mass spectrometry analysis, was appropriate or scientifically justified in this context.

Minor Comments:

1. Why were the cross-linked peptides further enriched via strong cation exchange chromatography?
2. More detail is needed on the infection conditions for all the viral infection experiments. What multiplicity of infections was used?
3. Information regarding the ICP4 antibody is missing.

Reviewer #3

(Remarks to the Author)

Bogdanow et al. describe the development of deep in-cell virus-host protein-protein interaction (PPI) profiling by combining proteome-wide crosslinking mass spectrometry (XL-MS) with selective enrichment of newly synthesized viral proteins, termed here Structural Host Virus Interactome Profiling (SHVIP). Bogdanow et al. observe over 700 PPIs within cells infected with herpes simplex virus type 1 (HSV-1), covering various intracellular compartments known to be involved in the viral life cycle. Several of these findings were validated by molecular genetics, and the predicted structures of a subset of HSV-1 - human complexes were described by combining the XL-MS data with AlphaFold-based structural modeling. According to the authors, SHVIP offers various conceptual advantages for virus-host PPI profiling and is in principle applicable for the study of native host cells infected with any virus inducing host shutoff. The method is described to be applicable for the in-depth characterization of viral gene functions during cellular infections.

Whereas the approach and the findings are interesting and novel, there are some shortcomings that I feel should be addressed.

Major concerns:

- The paper completely lacks an introduction into HSV-1, to help the reader to guide oneself in the findings and conclusions made. The authors state (line 90) that HSV-1 has a complex proteome; yet this is not described in this proteomics technological development manuscript. The authors are suggested to provide an overall schematic representation of the virion highlighting the structural proteins (can be a supplemental figure), their function and known host interactions (the latter two can be presented as a supplemental table instead if more convenient). The tables should also contain the same info for the non-structural proteins. The authors are additionally suggested to provide a more detailed view of the life cycle of HSV-1 than the one presented in Figure 2C, highlighting the key events, and known host-interactions (data prior to this manuscript) in order to improve the readability for the broader audience not experts in HSV-1 biology. The lack of these figures and this information makes the results and conclusions at times hard to follow. This is for instance highlighted in the paragraph on lines 235-255, and on lines 238-239 where the authors describe the selection of eight viral proteins for further study using AP-MS, but do not describe which proteins these are (only later by names in the text; however, these names lack an anchor point). Are these proteins for instance structural or non-structural, what are their known functions in the viral life-cycle of HSV-1 etc... The lack of an overall description of HSV-1 and its life-cycle is further an issue when reading lines 158-163 and 186-215, where the viral proteins are discussed, as well as on lines 175-177 when the viral life cycle is discussed.

- Figure 2a is too small to read to evaluate the data. Which community represents which color? Not being an expert on unsupervised clustering based on centrality indices, how does the data generate six communities in three cellular departments? Please elaborate in benefit of the broader audience. The authors state that "These data confirm that SHVIP can resolve virus-host PPIs across various cellular compartments." Does the data presented here reflect known interactions during the viral life cycle? As Figure 2a cannot be read due to its small size/small font, it is difficult to assess the validity of the findings.

- The authors map domain-specific host interaction partners to HSV-1 transmembrane proteins (lines 186-215), and state that "Overall, our data provide detailed, domain-level insights into virus-host interactomes at the endomembrane system, relevant for glycoprotein processing, membrane trafficking, adaptive immunity, egress and assembly." The way this passage is currently written, it is difficult to assess which identified interactions are novel, which are previously known and what the overlap between these two datasets are. What are the grey nodes in Figure 2e? The blue ones are apparently endoplasmic reticulum, the yellow ones RABs, and green ones HSV-1 tegument, and red HSV-1 membrane proteins. Is it possible in this figure to incorporate the distinction of novel vs. known interactions?

- The authors present a correlation between the AP-MS and SHVIP data in terms of identified interactors and claim a

significant overlap (Figure 3d and e). Whereas the p-value is significant, there are very few proteins belonging to both groups, and several that differ. It is generally accepted in the MS community that AP-MS data is noisy, and can capture unspecific preys, but can the authors comment on the noisiness of the SHVIP data? The authors state (lines 267-269) that “These results suggest that, while many PPIs are identified by both SHVIP and AP-MS, those depending on the in situ environment of the infected cell are uniquely captured by SHVIP.” Is there any evidence for this over the explanation of SHVIP capturing unspecific interactors/preys as well (i.e. is noisy), much as AP-MS?

- On lines 303-305 the authors state that “Only three inter-links (2 PPIs) could be mapped onto existing experimental structures of heterodimers (Figure 4a), reflecting the paucity of structural information on HSV-1-host interactions.” To me, this number seems very low in respect to the number of HSV-1 – host PPIs reported before in the manuscript, with 739 interactions involving 46 viral proteins (lines 157-158). Can the authors reflect on this low number and on how many experimental HSV-1 structures exist? This also correlates to the statement on lines 460-461.

- On lines 339-341 the authors state that “We obtained 68 models (18 involving viral proteins) that have both acceptable docking scores (pDockQ > 0.23) and an agreement of at least half of the inter-links with the prediction (Supplementary Table 4).” Is not the agreement of only half of the observed inter-links with the prediction a quite low agreement? Does the crosslinking data really support the structural models under these conditions? How was the crosslinking data filtered (for reliable hits)?

- The authors apply several different proteomics methods in their paper to address the HSV-1 – host interactome including SHVIP (i.e. labeling, XL-MS, and enrichment), AP-MS, co-IP and SILAC labeling. An overview presentation of these workflows and how they are connected in generating the various results and contributing to the different conclusions made in the manuscript would increase the understanding of the methodology for the broader audience.

- On lines 351-352 the authors state that “Two UL47-DDB1 cross-links were identified, with one of them satisfying the distance constraint in the corresponding AF2 model.” which is also highlighted in Figure 5b. There are only two crosslinks identified, one that supports the AF2 model, and one that does not. Can the authors comment on the reliability of the data? Is there any indication of flexibility in the regions involving the violated crosslink?

- The authors state on lines 461-462 that “Augmenting AF2 with SHVIP strengthened 18 protein complex predictions, which are potentially linked to DNA replication, immune evasion, and signaling.” I emphasize here the word “strengthened”. How many of these interactions/complexes are novel, i.e. identified based on the data presented in this manuscript, and how many interactions (not structure of complexes) were previously known?

Minor comments:

- The abbreviation (SHVIP) of the established workflow is not mentioned in the abstract and should be included there.

- On lines 52-54, the authors state that “However, AF has been developed mainly based on structural data of multimeric complexes within one organism, with co-evolutionary constraints likely different in host-pathogen scenarios.” References should be cited to support this statement.

- The reference to “Both challenges...” on line 77 is unclear; does it reflect the challenges presented on line 66-76 or the broader topics presented in the paragraphs on lines 43-57 (structural context and molecular determinants) or 58-76 (molecular environment). Please rephrase to improve readability and clarity.

- Please include more info and detail on the methodological setup for SHVIP, presented currently in Figure 1A, and please increase the image size/font size.

- Please verify that the text passage on lines 133-135 “Both the relative frequency and absolute numbers of PPIs involving viral proteins increased upon HPG enrichment (Supplementary Figure 1c) and showed reproducibility similar to previous XL-MS studies from complex samples 36 (Supplementary Figure 1d).” refers to the correct figure. Currently the Supplementary Figure 1c and d represent a comparison between Exp 1 and Exp2; not data from previous studies. Also, reading from the beginning of the manuscript Exp1 and Exp2 have not been described prior to this text passage.

- Please increase the image size and font size of Figure 1d. The edges connecting the nodes are very thin, and it difficult to verify that all human host proteins presented in the figure are in fact primary or secondary interactors. Also, black font on dark blue/purple is hard to see. Here, in this figure the HSV-1 proteins are mentioned, but there is not a more general presentation of the virus (please see major concern above) for the reader to connect these in terms of viral biology.

- Please clarify what the difference is between Figure 1d and Supplementary Figure 2a. Is the only difference that the human proteins are indicated in the latter? If so, please merge into one single figure only and increase the node font size.

- On lines 151-158 the authors describe that they combine data from two MS (MS2 and MS2/MS3) experiments to provide a high-coverage structural interactome of the infected cell. Please elaborate on the rationale for using the two methods in benefit of the broader audience; I assume that this is due to DSSO being MS-cleavable and the gain in data depth of combining these two approaches.

- Line 213-215, which are the five ER membrane proteins connected to the NEC?

- The line thickness between the two groups “XL within compartment” and “XL connecting compartments” are identical and cannot be discerned.
- Figure 4f does not in its current format show the violated crosslinks (>35 Å) (as indicated in the figure legend key).
- In Supplemental Figure 5a and Supplemental Figure 6a-c the contrast (choice of colors) makes it difficult to distinguish the residues highlighted in magenta. This choice of colors works better in Figure 6e where the residues in magenta are indicated in bolder lines.

Reviewer #4

(Remarks to the Author)

Version 1:

Reviewer comments:

Reviewer #1

(Remarks to the Author)

I appreciate the authors' efforts in addressing my comments. The paper is now in a more technically sound state in terms of modeling. However, I still have one major remaining concern regarding the AlphaFold version used in this study.

The authors made a notable effort to run AF2.3. However, I do not understand why they did not use AF3, as they should have had access to its stand-alone version during the revision period. AF3 runs significantly faster than AF2.3, and this paper could have provided a valuable platform for comparing its results with those of AF2.2 and AF2.3.

Additionally, AF2.3 is only a slightly improved version of AF2.2. The results would have differed more significantly if the authors had used AF2.1 (and 2.3 is not necessarily better than 2.1). In this regard, I recommend that the authors consult the following papers:

- <https://pubmed.ncbi.nlm.nih.gov/37548092/>
- <https://pubmed.ncbi.nlm.nih.gov/39528570/>

Given the innovative nature of this work, I believe it should align with the latest modeling technologies available in the field. Therefore, if the editor agrees, I suggest another round of revision, at the very least incorporating results from AF3.

Reviewer #2

(Remarks to the Author)

Based on the authors' revised manuscript and detailed response, all of this reviewer's initial concerns regarding the technical aspects of mass spectrometry analysis and specificity have been satisfactorily addressed. The authors have provided clear and comprehensive explanations regarding the specificity of their data, covering both the chemical and biological aspects of the cross-linking approach. The methodological rigor and the resulting dataset represent a significant advancement in the structural characterization of virus-host protein-protein interactions.

However, the authors did not adequately address Comment 1 from this reviewer. Demonstrating at least one instance of biological significance—specifically, a clear contribution to viral replication and/or pathogenesis—of one of the newly identified HSV-1–host interactions during viral infection would be important to support the study's validity. Unfortunately, the interaction between UL47 and UL48 presented in the Response to Referees Letter does not represent a newly identified HSV-1–host interaction but rather an HSV-1–HSV-1 interaction. Furthermore, this interaction is likely an unsuitable model for a novel interaction, as the intracellular binding of UL47 to UL48 has already been reported (PMID: 16014918) and the binding in HSV-2–infected cells has also been documented (PMID: 11087097). Additionally, a critical issue remains: the effects of the $\Delta 2-15$ and $\Delta 469-486$ mutations in UL48, which disrupt the binding between UL47 and UL48, are more pronounced than those observed with a null mutation in UL47. It has been reported that the deletion mutation in UL47 does not affect the expression level of UL48 (PMID: 8382306), nor does it reduce HSV-1 replication by more than 2 logs. Moreover, the series of experiments lacks appropriate controls (e.g., a repair virus).

Given these limitations, it remains questionable whether this methodology alone is not sufficient for publication in this journal.

Reviewer #3

(Remarks to the Author)

Bogdanow et al. describe the development of deep in-cell virus-host protein-protein interaction (PPI) profiling by combining proteome-wide cross-linking mass spectrometry (XL-MS) with selective enrichment of newly synthesized viral proteins;

Structural Host Virus Interactome Profiling or SHVIP.

I have carefully assessed the revised manuscript. The authors have addressed all my concerns and comments, and I have no further remarks on the manuscript.

Reviewer #4

(Remarks to the Author)

GENERAL REMARK:

We have renumbered all reviewer comments consecutively to make this document easier to navigate.

Point-by-point response to the reviewer's comments

We thank all four reviewers for carefully reading our manuscript and providing valuable feedback! We were pleased to read that our “elegant and sophisticated strategy” with “advanced methodology” (reviewer #2) resulted in “interesting and novel findings” (reviewer #3) and “a valuable set of interactions” (reviewer #1).

Further, following the reviewer's comments we now extensively revised our manuscript, including

- (i) a thorough re-analysis of proposed computational models, as requested by reviewer #1, which led to modified Figure panels 5f, 7b, 7d and Supplementary Figure panels 5a, 5b and 5c.*
- (ii) an explanation on specificity of the data in such types of experiments, as suggested by reviewer #2.*
- (iii) new Supplementary Figure panel 1e showing the agreement with literature data.*
- (iv) a new Figure 2, highlighting the HSV-1-host interactome as a separate Figure.*

Please refer to our point-by-point responses to the reviewers' as below.

Reviewer #1 (Remarks to the Author):

Bogdanow et al. investigate the interaction landscape of the HSV-1 infection cycle, effectively combining a specialized labeling and cross-linking strategy with structural modeling to propose viable human-virus complexes. The study presents a valuable set of interactions that could be leveraged by the scientific community to develop therapeutic components aimed at disrupting the HSV-1 infection cycle. However, before publication, the manuscript should be revised in light of the following computational improvements:

We thank the reviewer for the overall positive feedback and for raising important points!

- 1) - As the authors mention, human-viral and viral-viral protein complexes lack the typical coevolution signals utilized by AlphaFold. In such cases, extensive sampling has been shown to mitigate the limitations caused by shallow MSA profiles. Therefore, the authors should explain why they adhered to a single version of AlphaFold2 without experimenting with sampling parameters/weights to potentially improve the quality of their models. Furthermore, they should clarify why they did not use AlphaFold3 for these modeling tasks, which could offer an opportunity to enhance model quality over the current approach.

We thank the referee for these suggestions! We agree with the reviewer that our AF2.2 based pipeline is by now a bit outdated and therefore predicted all heterodimers with AlphaFold 2.3. We did not employ

AlphaFold3 because, at the time of our analysis (October 2024, source code was released mid-november 2024), it was only accessible via the AlphaFold server (<https://alphafoldserver.com/>), which imposed a limit of 20 predictions per day and required manual initiation. Using AF2.3 we now predicted each heterodimer with 25 models, and analyzed predicted models based on the AlphaFold confidence score ($0.8 \cdot \text{ipTM} + 0.2 \cdot \text{pTM}$) (see Supplementary table 4), replacing the pDockQ-based analysis we did in our initial version of the manuscript (please also see comment #3).

Then, we evaluated the agreement of cross-linking distance constraints in dependence of the model confidence (Figure 5e). We observed that models above a confidence score of 0.75 agreed well with the cross-linker distance constraint and can therefore be recognized as high confidence models. Additionally, a part of the models with lower confidence also agreed with the cross-linking data until a confidence score of 0.5 (Supplementary Table 4). As this includes also models that have experimental structures solved (gH-gL, UL30-UL42, NEC1-NEC2), we considered all models above 0.5 that agreed to at least 50% with the cross-linking distance constraints were shortlisted as valuable models.

Note that this analysis did not retrieve UL45-host protein interactions as major hits, due to differences in the AF2.2 pDockQ score and the AF2.3 based model confidence. Therefore, we removed UL45-TMED10 and UL45-TPBG predictions and replaced them by a different model that met our criteria (US12-PPIA). According to our new analysis, we updated Figure 4 (now shown as Figures 5), Supplementary Table 4 and Supplementary Figure 5.

*The reviewer correctly mentioned that virus-host PPIs are generally hard to predict using current AlphaFold pipelines. Indeed, this problem has been conceptually recognized by other researchers as well (Marrero et al., *Annu Rev Genomics Hum Genetics*, 2024, [10.1146/annurev-genom-120622-020615](https://doi.org/10.1146/annurev-genom-120622-020615); Bryant, *Curr Opin Struct Biol*, [10.1016/j.sbi.2023.102529](https://doi.org/10.1016/j.sbi.2023.102529) - now cited in the Introduction) and more recently also experimentally confirmed in a pre-print (<https://www.biorxiv.org/content/10.1101/2024.12.12.628104v1>), where out of 9,452 human-pathogen PPIs only 30 could be very well modeled. This indicates that accurately predicting virus-host PPIs requires a breakthrough in modeling pipelines that goes beyond sampling parameter/weight optimization. In line with this notion, we found no strong correlation between MSA depth and model quality (see our response to the reviewer's related comment below). This means mitigating shallow MSAs with extensive sampling might not give more accurate models.*

*Furthermore, even if the overall model quality may be improved by computational innovations, it would still be essential to biologically validate the models with experimental data. Therefore, we chose to focus our work **not** on solving the problem of current prediction pipelines for host-pathogen PPIs but on generating host-virus cross-linking data at a systems level in intact infected cells. We have shown that the cross-links from our SHVIP method can directly support the validity of host-virus PPIs and indicate an appropriate model confidence score cutoff (see above), confirming that our experimentally guided approach offers an efficient way to identify relevant host-pathogen PPI models.*

- 2) - While pDockQ is a useful metric for predicting complex quality, its familiarity is largely limited to the computational biology community. To improve the accessibility and clarity of their results, the authors are encouraged to also report confidence scores, as well ipTM or iPAE, which are more widely recognized and understood across the broader scientific community.

We agree with the reviewer that confidence scores are more widely recognized, especially for AlphaFold 2.3 since the original pDockQ score was only trained for AF2.2. As mentioned in comment #1, we performed a new round of predictions with AF2.3 and re-analyzed our data according to confidence scores (derived from ipTM and pTM scores). Accordingly, we substituted pDockQ scores in Figures 4 and 6 as well as in the text to confidence scores and now we report pTM, ipTM and confidence score for each model in Supplementary table 4.

We also included predicted alignment error (PAE) plots for the selected high-scoring models shown in Figure 4 (now shown as Figure 5) to illustrate model quality of high-confidence predicted models. Please note that we did not include PAE plots in figures 6 and 7. These figures focus on the discovery of linear interaction motifs in poorly-structured regions, the analysis of which is based on pLDDT scores.

- 3) - Additionally, the authors are advised to provide an analysis of Neff (the effective number of sequences in the MSA) versus the predicted quality of the models. This would offer insights into how the depth of the MSA impacts the quality of the generated models, providing a clearer understanding of the relationship between sequence diversity and model accuracy.

Following the reviewer's suggestion, we analyzed the effective number of sequences in the MSA (Neff) versus AlphaFold confidence scores of the predicted models. We did not observe strong correlation between higher Neff and higher confidence of the predicted model. In fact, many host-virus dimers have acceptable, or even good confidence scores despite having low Neff. Likewise, some host-host dimers have low confidence scores but high Neff, showing Neff may not be a good indicator for model confidence.

Reviewer Figure 1: Scatterplot of dimeric protein structure prediction with AlphaFold 2.3 comparing AlphaFold confidence score against the effective number of sequences used in the multiple sequence alignment (Neff).

Reviewer #2 (Remarks to the Author):

In this paper, the authors present an innovative approach for comprehensively detecting virus-host protein-protein interactions (PPIs) without cell lysis in the affinity purification of protein complexes, which is a conflicting procedure to detect PPIs. The combination of crosslinking mass spectrometry and selective bio-orthogonal labeling of viral proteins represents an elegant and sophisticated strategy. Using this newly developed method, the authors identified more than 700 PPIs in HSV-1-infected cells. Furthermore, their analyses of cross-linked peptide data provided domain-specific insights into PPIs, potentially improving the accuracy of predicting interactions between structured and disordered regions. These data will contribute significantly to advancing our understanding of HSV-1 infection. The concept of this study is intriguing and can be easily applied to other viruses. The authors present cutting-edge data supported by advanced technology.

On the other hand, there seem to be concerns regarding the comprehensiveness and specificity of the data presented. In addition, this study did not thoroughly evaluate the biological significance of the newly identified HSV-1-host interactions in the context of viral infection. Such data will greatly strengthen the results of this study and encourage researchers in the broader virology research community to utilize this newly developed approach.

We thank the reviewer for the positive comment and for raising important points, which we address in response to their specific comments below.

Major Comments:

- 4) The authors should further investigate the biological significance (that is, contribution to viral replication and/or pathogenesis) of at least one of the newly identified HSV-1-host interactions in the context of viral infection.

While we agree with the reviewer that our dataset provides a treasure trove of host-virus PPI candidates that would be of interest for functional follow-up studies, the focus of our manuscript is on method development and resource generation. Therefore, we feel that a deep functional interrogation of our dataset is outside the scope of this manuscript.

Having said this, we heeded the reviewer's advice and followed up on the UL47-UL48 interaction, which was supported by six inter-links between both proteins (see Reviewer Figure R2a). UL48 is also known as alpha-TIF (trans-inducing factor) or VP16 and is an important viral protein as it makes part of the viral particle and activates the transcription of the first viral genes (immediate-early genes) following entry. While early studies have proposed a regulatory relationship between UL47 and UL48 (see McKnight et al., 1987, <https://doi.org/10.1128/jvi.61.4.992-1001.1987>), a physical interaction between both proteins has not yet been demonstrated in infected cells.

Our AlphaFold predictions of UL47-UL48 provided two mutually exclusive structural models (see Reviewer Figure 2b,d). One model shows the N-terminal disordered region of UL48 (aa2-15) forming an alpha-helix that binds to a pocket of UL47; the other model shows the same structure and UL47 binding site for the C-terminal disordered region of UL48 (aa 469-486). To test if any of the models correctly identified the binding interface, we created UL48 deletion constructs lacking either the N- or C-terminal helical region (Δ 2-15, Δ 469-486). Then we performed AP-MS experiments with wild-type and mutant proteins from transfected cells in the presence of co-transfected HA-tagged UL47, which was used as bait (Reviewer Figure 2c,e). We found that both AlphaFold-proposed interaction sites contribute to precipitating UL47 from cell lysates, but the c-terminal deletion had a much stronger effect. When we integrated these mutations into the viral genome, we observed small growth defects for viruses lacking either the N- or C-terminal region in UL48 and a synergistic growth defect for viruses lacking both regions (decrease in viral multiplication by \sim 2 orders of magnitude, Reviewer Figure 2f). An immunoblot analysis of UL47 and UL48 protein abundance kinetics across the infectious cycle, revealed that UL48 was present at much lower levels in the mutant viruses (Reviewer Figure 2g). Similar growth defects have been observed in mutant viruses deleted for the entire UL48 gene (Weinheimer et al., J Virol, 1992; [10.1128/jvi.66.1.258-269.1992](https://doi.org/10.1128/jvi.66.1.258-269.1992)).

While these results demonstrate an interaction between UL48 and UL47 and show a critical role of the c-terminal region in mediating this interaction, the effect of the interaction on the growth of the virus is harder to interpret. This is because of the following reasons:

1. The C-terminal region in UL48 HSV-1 that we mutated is not only the interaction interface for UL47, but is also a part of the H2 part of the VP16 transactivation domain (Hirai et al., *Int J Dev Biol*, 2012, [10.1387/ijdb.103194hh](https://doi.org/10.1387/ijdb.103194hh)). Therefore, we cannot exclude that growth defects of viruses lacking the UL48 C-terminus are caused by transcriptional effects.
2. The mutant viruses had defects in the accumulation of UL48, which was very surprising, as destabilization of UL48 has not yet been observed. This destabilization may be a result of disrupting interaction with UL47 but it may also be interaction-independent (e.g. a direct consequence of the mutations).

Reviewer Figure 2: Interaction between viral tegument proteins UL47 and UL48, as detected by SHVIP.

a Depiction of the cross-links between UL48 and UL47 and within UL47. **b** Best ranked AlphaFold2.3 multimer model (rank 0), showing UL47 in surface mode in grey and UL48 as cartoon colored according to pLDDT score. An N-terminal proposed interaction motif is highlighted. **c** Volcano plot of AP-MS experiment of cells co-transfected with HA-tagged UL47 and UL48 wild-type (WT) or a UL48 mutant where amino acids 2-15 were deleted. HA-UL47 was precipitated. UL47 and UL48 are highlighted. **d** AlphaFold2.3 multimer model rank 1, showing UL47 in surface mode in grey and UL48 as cartoon colored according to pLDDT score. A C-terminal proposed interaction motif is highlighted. **e** Volcano plot of AP-MS experiment of cells co-transfected with HA-tagged UL47 and UL48-WT or a UL48 mutant where

amino acids 469-486 were deleted. HA-UL47 was precipitated. UL47 and UL48 are highlighted. f Growth kinetics of WT HSV-1 compared to HSV-1 containing N-terminal-, C-terminal- and double deletion mutant of UL48. Viral growth was measured by determining infectious units (IU) per mL cell culture supernatant. g Western blot analysis performed on cell lysates from HSV-1-infected cells containing HA-tagged UL47 with either UL48-WT, the N-terminal UL48 deletion mutant, the C-terminal deletion mutant, or the double deletion mutant. Cells were harvested at different time points after infection according to growth kinetics measurements in f.

In conclusion, our results provide preliminary evidence for a UL47-controlled regulation of the central viral transcriptional regulator UL48, showing how SHVIP can generate valuable leads for functional studies. However, we feel that the current data are too premature to include them in the manuscript. Thoroughly validating the role of the UL47-UL48 interaction will require substantially more experiments with more specific virus mutants, which are not feasible in a timely manner.

In our view, the most exciting aspect of our work is that we have developed the first method that allows the systematic structural characterization of virus-host PPIs within intact infected cells. We have used this approach to generate a resource of PPIs and structural models, several of which were validated by mutational analysis. We would prefer to keep our manuscript focused on these points. We feel that Nature Communications is an excellent home for such a method and resource- focused study, because this journal has published many papers of similar scope in recent years – both in the virus field (e.g. Soh et al., Nat Commun 2024, 10.1038/s41467-024-54668-2; Fossati et al., Nat Commun 2023, 10.1038/s41467-023-40724-w) and the structural proteomics field (e.g. Chen et al., Nat Commun 2023, 10.1038/s41467-023-39485-3, Michael et al., Nat Commun 2024, 10.1038/s41467-024-52844-y).

- 5) The authors stated, "Together, this indicates that SHVIP sensitively captures a large part of the available viral proteome." Given that HSV-1 expresses approximately 100 viral proteins in infected cells, detecting only 46 HSV-1 proteins in this dataset does not convincingly demonstrate the coverage of a substantial portion of the viral proteome. The interactions between HSV-1 and host proteins are likely far more complex, and this study may have uncovered only a small fraction of these interactions.

In this study we used the HSV-1 reference proteome (TaxonID 10299) downloaded from Uniprot including 73 protein sequences. Of those, 68 proteins could be identified in the HPG-enriched shotgun proteomics experiments, with 60 proteins consistently identified across all 4 replicates (see Supplementary table 1). XL-MS is known to have lower sensitivity than proteomics, as XL-MS targets the much lower abundant cross-links compared to the large excess of linear peptides (Steigenberger B., et al., ACS Cent. Sci. 2019, 5, 9, 1514–1522). Using SHVIP, we detected 46 cross-linked viral proteins, i.e. 77% of the consistently identified HSV-1 proteome, which is a remarkable achievement for XL-MS. Furthermore, the distribution of intensity values (intensity based absolute quantification, iBAQ-values) of cross-linked proteins shows that we could detect cross-links across the available proteome, also involving low-abundant proteins

(Reviewer Figure 3). Thus, we think it's fair to say that we could detect cross-links for most of the MS detectable viral proteome.

Reviewer Figure 3. Comparison of iBAQ values of proteins detected in HPG-enriched sample (see Supplementary table 1) against their corresponding rank sorted after iBAQ with proteins found cross-linked in our PPI map highlighted.

Nevertheless, following the reviewer's suggestion, we toned down to the following sentence in the revised manuscript, "Together, this indicates that SHVIP captures many viral proteins from non-structural and structural categories."

- 6) The authors must explain more clearly how they ensured the specificity of their data. Although crosslinkers are designed to react with specific amino acids, nonspecific binding can still occur, potentially resulting in false positives. It is crucial to specify the extent to which false positives were accounted for, and how they were addressed or excluded from the analysis.

We agree with the reviewer that it's very important to address the specificity of the data. Indeed, we made a great effort to control the specificity in three aspects: (1) the specific matching of MS spectra to identification of cross-linked peptides, (2) the biological specificity of the identified cross-links, and (3) the chemical specificity of the cross-linker mentioned by the reviewer.

(1) Specific matching of MS spectra to cross-linked peptides

Controlling false spectral matches in XL-MS is an important issue (see also our recent publication on this topic: Bogdanow et al., Mol Syst Biol, 2024, <https://doi.org/10.1038/s44320-024-00079-w>). In this manuscript (see Figure 1), we showed that the FDR of inter-link matches can be properly controlled using the target-decoy competition method, which was used to analyze the cross-linking data in this study. Furthermore, we incorporated two additional criteria to stringently control the FDR of inter-protein cross-links: 1) we used separate FDR control for inter-protein and intra-protein cross-links, which was shown to better control the FDR of inter-links, and 2) in addition to the residue-pair FDR, we also controlled the FDR at the PPI level. The benefits of these two additional criteria have been described previously (Fischer et al., Anal Chem 2017, <https://doi.org/10.1021/acs.analchem.6b03745>, Leitner et al., Structure 2020, <https://doi.org/10.1016/j.str.2020.09.011>).

We explain this procedure in the methods section of the manuscript and added clarifications upon revision (underlined)

“Raw files from both experiments (MS2-only and MS2-MS3 acquisition) were searched separately, results were then combined and the FDR at 1% was applied at the level of residue-residue connections, separately for intra- and inter-links on the basis of a target-decoy competition strategy using randomized decoys. Following this, all cross-links were retained when they matched to a viral protein or to host protein with a viral interaction partner (virus-centric network). This network still contains host-host cross-link between host proteins that are also linked to viral proteins. The PPI-level FDR in this set of PPIs was assessed based on the ratio of decoys to targets in the virus-centric network.”

(2) Biological specificity of the identified cross-links

The biological specificity of the data is underscored by the following observations:

- (1) We performed cross-linking on intact cells, where the cellular membrane system stays intact. We therefore expect the cross-links are subcompartment specific. Reassuringly, 99.6% of the identified cross-links comply with this restraint (see Figure 3d).
- (2) When mapping cross-links on existing high-resolution structures, we showed the cross-links were in excellent agreement with the structural data (e.g., Figure 5a, UL30-UL42, gH-gL, US11-TAP1).
- (3) Cross-linked host proteins show the known bi-modal distribution (see Burke et al., 2023; <https://doi.org/10.1038/s41594-022-00910-8>) of confidence scores in AF2 models (with ~½ excellent host-host models), underscoring the validity of the data at the host-host level (see Figure 5c).
- (4) In response to reviewer #3, major comment #3, we added new data in the revised manuscript by providing a comparison of our cross-linking data to functional studies in the HSV-1 virology field.

In this comparison, we showed that ~ 23 % of viral-host cross-links involving viral proteins have literature evidences to support their interactions (see Supplementary Figure 1e), indicating that our cross-linking MS method captures functional viral-host interactions.

(3) Chemical specificity of the cross-linker

The reviewer is correct that no cross-linker has perfect specificity. The DSSO cross-linker used here reacts with primary amines on lysine side chains and, to a lower extent with hydroxyl-bearing side chains from serines or threonines (see Cao et al., <https://doi.org/10.1021/acs.jproteome.3c00037>). Some workflows include serine and threonine cross-linking during database search. However, this increases the search space, giving more opportunities for wrong matches which may result in fewer overall identifications (as the sensitivity is controlled by accepting at most 1 % false matches). Therefore, we did not include serine and threonine cross-linking in our searches, aiming to maximize the identification of the chemically most likely lysine cross-links.

- (7) Crosslinking between peptides generates irregular fragmentation patterns, making the mass spectra of cross-linked peptides significantly more challenging to analyze than those of regular peptides. Furthermore, the presence of various post-translational modifications in HSV-1-infected cells complicates the identification of crosslinked peptides and determination of crosslinking sites. Given these complexities, it is reasonable to question whether a standard FDR threshold of <1%, typically used in conventional mass spectrometry analysis, was appropriate or scientifically justified in this context.

While it is correct that cross-linking generates more complex (but not irregular) fragmentation pattern and they are therefore also more challenging to analyze than regular peptides, it is established that the FDR approaches used by the community can accurately control the error rates at the spectrum, cross-link and PPI-level (Leitner et al., Structure 2020, <https://doi.org/10.1016/j.str.2020.09.011>).

The reviewer is correct that the presence of post-translational modification can lead to systematic errors in shotgun proteomics (see Bogdanow et al., 2016, Mol Cell Proteomics, [10.1074/mcp.M115.055103](https://doi.org/10.1074/mcp.M115.055103)). However, there are computational methods based on target-decoy competition that handle such systematic errors (Bogdanow et al., 2016, Mol Cell Proteomics, [10.1074/mcp.M115.055103](https://doi.org/10.1074/mcp.M115.055103), Bogdanow et al, 2024, Mol Syst Biol) in both shotgun proteomics and XL-MS. Furthermore, we disagree that the presence of an unknown PTM site complicates the determination of the XL-site. If there is an unknown PTM on a peptide, this peptide will have a different precursor mass than the unmodified peptide. This mass difference allows the search algorithm to easily distinguish the peptide with the PTM and the unmodified variant. This does not interfere with the determination of the XL-site and the sequence of the cross-linked peptide.

The 1 % FDR cut-off is standard for XL-MS studies conducted on uninfected human/mammalian samples. We consider this cut-off appropriate for our study of HSV-1 infected cells which have a similar PTM complexity compared to uninfected cells. For more in-depth analysis on FDR in XL-MS, please see: Lenz et al., 2021, Nat Commun: <https://doi.org/10.1038/s41467-021-23666-z>,

Bogdanow et al., 2024, MSB: <https://doi.org/10.1038/s44320-024-00079-w>.

Importantly, we make all RAW data available through PRIDE, which will give others the opportunity to re-analyze the data with different FDR settings and/or inclusion of PTMs as variable modifications.

Minor Comments:

- (8) Why were the cross-linked peptides further enriched via strong cation exchange chromatography?

Cross-linked peptides are typically less abundant than non-cross-linked peptides, which makes it necessary to further enrich the cross-linked peptides to increase the sensitivity of cross-link detection. Cross-linked peptides can be enriched based on charge (because they have two tryptic sites, they have higher charges, on average, than non-cross-linked peptides) using strong cation exchange (SCX). In this case, the bulk of the non-cross-linked peptides carries a charge of +2/+3, and elutes early in SCX, while the higher charged peptides (+4, +5...) elute late in the chromatogram. The later fractions are those that are measured in the mass spectrometer.

- (9) More detail is needed on the infection conditions for all the viral infection experiments. What multiplicity of infections was used?

For all infection experiments, we used a multiplicity of infection of 5 IU/cell. We added this information in the "Cells and viruses" paragraph of the Material and Methods section.

- (10) Information regarding the ICP4 antibody is missing.

For immunotitration, we used a 1:200 dilution of the mouse anti-ICP4 antibody clone H943 from Santa Cruz Biotechnology (sc-69809). We added this information in the "Cells and viruses" paragraph of the Material and Methods section.

Reviewer #3 (Remarks to the Author):

Bogdanow et al. describe the development of deep in-cell virus-host protein-protein interaction (PPI) profiling by combining proteome-wide cross-linking mass spectrometry (XL-MS) with selective enrichment of newly synthesized viral proteins, termed here Structural Host Virus Interactome Profiling (SHVIP). Bogdanow et al. observe over 700 PPIs within cells infected with herpes simplex virus type 1 (HSV-1), covering various intracellular compartments known to be involved in the viral life cycle. Several of these findings were validated by molecular genetics, and the predicted structures of a subset of HSV-1 - human complexes were described by combining the XL-MS data with AlphaFold-based structural modeling. According to the authors, SHVIP offers various conceptual advantages for virus-host PPI profiling and is in principle applicable for the study of native host cells infected with any virus inducing host shutoff. The method is described to be applicable for the in-depth characterization of viral gene functions during cellular infections.

Whereas the approach and the findings are interesting and novel, there are some shortcomings that I feel should be addressed.

We thank the reviewer for her/his overall positive evaluation!

Major concerns:

(11)- The paper completely lacks an introduction into HSV-1, to help the reader to guide oneself in the findings and conclusions made. The authors state (line 90) that HSV-1 has a complex proteome; yet this is not described in this proteomics technological development manuscript. The authors are suggested to provide an overall schematic representation of the virion highlighting the structural proteins (can be a supplemental figure), their function and known host interactions (the latter two can be presented as a supplemental table instead if more convenient). The tables should also contain the same info for the non-structural proteins. The authors are additionally suggested to provide a more detailed view of the life cycle of HSV-1 than the one presented in Figure 2C, highlighting the key events, and known host-interactions (data prior to this manuscript) in order to improve the readability for the broader audience not experts in HSV-1 biology. The lack of these figures and this information makes the results and conclusions at times hard to follow. This is for instance highlighted in the paragraph on lines 235-255, and on lines 238-239 where the authors describe the selection of eight viral proteins for further study using AP-MS, but do not describe which proteins these are (only later by names in the text; however, these names lack an anchor point). Are these proteins for instance structural or non-structural, what are their known functions in the viral life-cycle of HSV-1 etc... The lack of an overall description of HSV-1 and its life-cycle is further an issue when reading lines 158-163 and 186-215, where the viral proteins are discussed, as well as on lines 175-177 when the viral life cycle is discussed.

We thank the reviewer for highlighting potential sources of confusion. To make the manuscript more understandable for the general audience, we now included a paragraph about HSV-1 in the Results section as below:

“We applied this methodology to HSV-1 infected human embryonic lung fibroblasts (HELFs). Fibroblasts support the HSV-1 lytic replication cycle, which is characterized by a temporal cascade of viral gene expression and the shutoff of host protein synthesis 46. The host shutoff is achieved by a combination of transcriptional and post-transcriptional mechanisms that are installed within the first 3-6 hours after infection by the concerted action of ICP4, ICP22, ICP27 and the viral endonuclease VHS/UL41. After this early phase of infection, HSV-1 - like all herpesviruses - has a nuclear stage of viral DNA replication and capsid assembly, and a cytoplasmic stage in which the outer tegument and membrane envelope layers of the newly forming virus particles are assembled. The peak of virus progeny production and release is typically reached around 24 h post infection. Based on this timeline, we decided to label the cells with HPG from 7 to 24 hours post infection.”

Additionally, we present a schematic of the virion explaining the different classes of HSV-1 proteins in the new Figure 2a.

(12)- Figure 2a is too small to read to evaluate the data. Which community represents which color? Not being an expert on unsupervised clustering based on centrality indices, how does the data generate six communities in three cellular departments? Please elaborate in benefit of the broader audience. The authors state that “These data confirm that SHVIP can resolve virus-host PPIs across various cellular compartments.” Does the data presented here reflect known interactions during the viral life cycle? As Figure 2a cannot be read due to its small size/small font, it is difficult to assess the validity of the findings.

Figure 2a (revised Figure 3a) represents the interactome data from revised Figure 2, with the color coding reflecting the different communities, as indicated by the colored numbers next to the colored protein communities. The host proteins within the different communities were subjected to GO analysis, showing that the communities are enriched with GO terms specific to certain subcellular compartments.

This community clustering, first published in a seminal paper by Girvan and Newman, PNAS, 2002, aims to identify communities of nodes that have a high degree of connections between them. A good introductory may be the corresponding wikipedia page:

https://en.wikipedia.org/wiki/Girvan%E2%80%93Newman_algorithm

The algorithm first finds those edges through which many of the shortest paths between pairs of nodes run (“edge betweenness”). It then progressively removes these edges from the original network and the remaining connected nodes represent the communities. Thus, a community has a higher degree of connectivity within itself than to proteins from other communities.

We reason that the proximity-based nature of the cross-linking data should therefore capture tightly connected communities based on the compartmental localizations of proteins and their interactors. Indeed, based on GO analysis we observed that the six communities defined by edge betweenness clustering reflect. Identifying more communities than cellular compartments is not unusual because the communities may represent compartments as well as large protein complexes (such as ribosomes and nucleosomes in our analysis).

We have used related bioinformatic approaches (XL-data for spatial mapping) in previous publications (Bogdanow et al., Nat Microbiol, 2023, <https://doi.org/10.1038/s41564-023-01433-8>; Zhu et al., Nat Commun, 2024, <https://doi.org/10.1038/s41467-024-47569-x>)

We removed the protein labels in revised Figure 3a, as they were hardly visible and layout of the network is identical to the one displayed in Figure 2a. We added this information to the legend of revised Figure 3a.

Does the data presented here reflect known interactions during the viral life cycle?

Yes, some of the data represent known interactions during the viral life cycle. We now provide the reviewer with a list of interactors and which ones are known from the literature. Please see Supplementary table 2 for a detailed list of PPIs that are supported by literature evidence from focused experiments (excluding large scale studies, such as Y2H). The data is graphically illustrated in Supplementary Figure 1e. We refer to it in the results section:

“Published focused biochemical studies confirm ~23 % of the detected PPIs from infected cells (Supplementary Table 2, Supplementary Figure 1e), suggesting that our network captures functionally relevant interactions.”

- (13)- The authors map domain-specific host interaction partners to HSV-1 transmembrane proteins (lines 186-215), and state that “Overall, our data provide detailed, domain-level insights into virus-host interactomes at the endomembrane system, relevant for glycoprotein processing, membrane trafficking, adaptive immunity, egress and assembly.” The way this passage is currently written, it is difficult to assess which identified interactions are novel, which are previously known and what the overlap between these two datasets are. What are the grey nodes in Figure 2e? The blue ones are apparently endoplasmic reticulum, the yellow ones RABs, and green ones HSV-1 tegument, and red HSV-1 membrane proteins. Is it possible in this figure to incorporate the distinction of novel vs. known interactions?

We thank the reviewer for the comment. Our manuscript is the first large-scale proteomic characterization of HSV-1 host interactions and there is no other dataset on known or curated interactions. However, to still give the readers an indication on which interactions are already known, we checked the available literature on HSV-1- host and HSV-1-HSV-1 interactions for biochemical evidence in from infected cells. We now present this information in Supplementary table 2.

In addition, we modified Figure 3e by adding bold outlines to all soluble proteins that have been previously observed to interact with any of the HSV-1 membrane proteins. The grey nodes in this figure are host proteins that are not annotated as ER proteins or RABs, or viral proteins that are not tegument proteins or transmembrane proteins. We more clearly depict this now in the figure panel.

Additionally, we now reference the known PPIs when referencing the Figure panel 3e to clarify which interactions have been previously observed.

“We mapped the domain-specific interaction partners onto HSV-1 transmembrane proteins, which validated 23 known PPIs (Figure 3e, Supplementary Figure 2a, Supplementary Table 2).”

- (14)- The authors present a correlation between the AP-MS and SHVIP data in terms of identified interactors and claim a significant overlap (Figure 3d and e). Whereas the p-value is significant, there are very few proteins belonging to both groups, and several that differ. It is generally

accepted in the MS community that AP-MS data is noisy, and can capture unspecific preys, but can the authors comment on the noisiness of the SHVIP data? The authors state (lines 267-269) that “These results suggest that, while many PPIs are identified by both SHVIP and AP-MS, those depending on the *in situ* environment of the infected cell are uniquely captured by SHVIP.” Is there any evidence for this over the explanation of SHVIP capturing unspecific interactors/preys as well (i.e. is noisy), much as AP-MS?

We also believe this is a very important question and this is one of the reasons we set out to perform a systematic comparison of AP-MS and XL-MS results. Much of the noise in AP-MS arises from the destruction of the native cellular context, which is very likely not an issue in SHVIP because we performed cross-linking directly on intact cells. Further, AP-MS will identify proteins non-specifically bound to the enrichment beads, which cannot be completely eliminated by comparing to negative controls. XL-MS identifies only direct protein-protein interactions, which precludes the capture of non-specific binders. Nevertheless, there are potential sources of unspecificity in XL-MS, which are discussed in detail in our response to point #6 of reviewer 2.

Additionally, to assess whether our XL-MS data produces noisy identifications we analyzed the experiment-to-experiment reproducibility in the XL-MS data. We observed only minor differences in experiment-to-experiment reproducibility for AP-MS supported SHVIP identifications (41.7 % overlap at fold-change log2 cut-off of 4) compared to SHVIP identifications that are not supported by AP-MS (39.7 % overlap), which could be an indication that PPIs uniquely identified by XL-MS are not inherently more noisy than PPIs found with both methods (Reviewer Figure 4, see below).

Reviewer Figure 4: Detected PPIs involving viral proteins in the two experiments of cross-linked and HPG-enriched samples in dependency of whether or not the PPIs have also been detected in AP-MS experiments using 4 as a log2 fold change cutoff.

The reviewer also pointed out that the overlap of XL-MS and AP-MS is modest. We think one important reason for this discrepancy is that XL-MS captures protein interactions in their cellular context, and thus is favorable for weak or context-dependent interactions, such as the ones listed below:

- *UL49 (see Figure 4 g/h and associated text), which has demonstrated the ability to associate with histones in experiments using the highly conserved alpha-herpesvirus ortholog from BoHV-1 (Ren et al., 2001, J Virol, [10.1128/jvi.75.17.8251-8258.2001](https://doi.org/10.1128/jvi.75.17.8251-8258.2001)) and undergo liquid-liquid phase separation (LLPS, see Xu et al., 2021, Mol Cell, [10.1016/j.molcel.2021.05.002](https://doi.org/10.1016/j.molcel.2021.05.002)). They are*

identified in XL-MS but not in AP-MS. This indicates interactions in the cellular context of liquid-liquid phase separation may be missed by AP-MS but can be captured by XL-MS.

- UL12 interactors, into which we investigated in the manuscript. As shown below and in Supplementary Figure 3 a of the manuscript, HA-UL12 was purified from infected cells and compared to anti-HA control purifications using the WT virus (Reviewer Figure 2). We found that many cross-linking partners of UL12, such as the MAPK8 protein and 14-3-3 proteins only had a weak tendency to co-enrich with the bait compared to the background in our experimental setup. However, we have demonstrated in Figure 7 of the manuscript using mutational approaches that those XL-MS based hits represent genuine interactions (new Figure 7 e,f), even though they were not identified as candidate interaction partners in the initial AP-MS experiment with UL12.

Reviewer Figure 5: Volcano plot of AP-MS results of HA-tagged UL12 in context of HSV-1 infected cells, as part of manuscript Supplementary Figure 3a. Cross-linked UL12 interaction partners are highlighted as blue dots. This includes 14-3-3-proteins and MAPK8, which have been further validated as UL12 interaction partners in our manuscript by site-directed mutagenesis. The two dotted lines correspond to the two cut-offs at log₂ fold change of 2 and 4 respectively.

Further, inspired by the reviewer's comment, we now discuss the discrepancies of AP-MS and XL-MS in more detail in the manuscript (underlining indicates passages that were integrated upon revision).

“SHVIP can capture weak and context-sensitive PPIs because it capitalizes on protein crosslinking within intact cells. Such interactions may be disrupted during cell lyses and therefore elusive to AP-MS. While SHVIP allows characterizing viral proteins that are challenging to study outside their cellular context (such as proteins undergoing LLPS⁸⁵ or transmembrane proteins⁸⁶), the

sensitivity of XL-MS workflows depends a lot on the abundance and structural complexity of the specific protein. Although our approach could enrich viral proteins, it does not eliminate identification bias originated from protein abundances. When interpreting XL-MS data, it is therefore critical to keep in mind that interactions for abundant proteins are more likely to be covered compared to low-abundant proteins. This abundance bias is reduced in AP-MS, which because detecting linear peptides and focusing on interactors of one individual protein may increase sensitivity. Beyond sensitivity considerations, AP-MS is primarily based on protein affinity whereas XL-MS is driven by protein proximity. These methodological differences likely explain why both techniques provide overlapping information while also offering a substantial degree of complementarity (Figure 4).

XL-MS captures a snapshot of the in situ occurring interactome that preserves the context of higher-tier interactors. It is therefore better able to capture PPIs only existing in specific subcellular contexts, such as LLPS-based condensates. For instance, we identified UL49 as the most connected viral protein in the XL-MS-based interaction network, which is in agreement with previous findings that UL49 interacts with a variety of viral and cellular proteins^{61,63,64,87}, such as histones⁶⁶⁻⁶⁸. It is possible that UL49 achieves the organization of many PPIs through its ability to undergo LLPS²⁰.

(15)- On lines 303-305 the authors state that “Only three inter-links (2 PPIs) could be mapped onto existing experimental structures of heterodimers (Figure 4a), reflecting the paucity of structural information on HSV-1-host interactions.” To me, this number seems very low in respect to the number of HSV-1 – host PPIs reported before in the manuscript, with 739 interactions involving 46 viral proteins (lines 157-158). Can the authors reflect on this low number and on how many experimental HSV-1 structures exists? This also correlates to the statement on lines 460-461.

We thank the referee for this comment! During review of our manuscript, we realized that we indeed missed an interaction: UL42-UL30 (Gustavsson et al., NAR, [10.1093/nar/qkae374](https://doi.org/10.1093/nar/qkae374)). This brings the count of structure-validated cross-links to a total number of 5 inter-links and 3 PPIs. We included the UL30-UL42 interlinks in Figure 5a.

The low number of interactions with solved structures can be explained as follows:

We found 38 pdb entries of HSV-1 multimeric structures (virus-host or virus-virus, see Reviewer Table 1), of which 18 are non-redundant in terms of protein identity and sequence coverage. Of those 18, four are antibody-glycoprotein complexes, which cannot be observed in our cell-culture based system as no antibody is present. Further, three are between glycoproteins and cell surface receptors. These interactions occur during entry of the virus into the cell, whereas we performed cross-linking at a late infection stage. Thus, there are only 11 solved heteromultimeric structures involving HSV-1 proteins that we may have possibly captured in SHVIP (highlighted bold in Reviewer Table 1).

Most of those 11 structures are not compatible with our cross-linking data because 1) they contain no or very few lysines (XL reaction site), 2) the cross-linked residues are not resolved in the structure, 3) one or

more proteins was not detectable in our sample according to standard shotgun proteomics. There are only two complexes (PDB code: 6T5A, 6M5R) that have remained undetected for unknown reasons, which we consider reasonable in such a proteome-wide XL-MS experiment.

Reviewer Table 1: Heteromultimeric structures in the protein data bank involving HSV-1 only or HSV-1 and human proteins. Non-redundant multimers excluding antibody-antigen structures and structures involving cell surface receptors are highlighted by bold font. To compile this list, we used the Advanced Search Query Builder tool available on the RCSB-PDB online resource. We specified search parameters “Source Organism Taxonomy Name (Full Lineage)” to match “Human alphaherpesvirus 1” and “Number of Distinct Protein Entities” to be greater than 1 to receive multimeric structures. Additionally we filtered out all models that contained proteins of different origin than HSV-1 or human as well as homodimers. The entry 4XAL was removed as one of the two entities was not assigned. Note that gH-gL heterodimer, which we depicted in Figure 5a is a structure from related HSV-2 and does therefore not occur in this table. The table is alphabetically/numerically sorted by pdb entry.

pdb	proteins	comment	PPI detected (yes/no)	# plottable interlinks
1DML	UL42, UL30	Redundant info as in 9ENP	yes	0
1JMA	gD, TNFRSF14	PPI with cell surface receptor for entry	no	0
2GJ7	gE, IgG	Interaction with antibody	no	0
2PHE	ul48, Sub1	sub1 not detected with shotgun proteomics in enriched samples	no	0
2PHG	UL48, Tfiib	Tfiib not detected with shotgun proteomics in enriched samples	no	0
3SKU	gD, PVRL1	PPI with cell surface receptor for entry	no	0
3U82	gD, PVRL1	redundant info to 3SKU	no	0
3WV0	gB, PILRA	PPI with cell surface receptor for entry	no	0
4WPH	icp0, USP7	USP7 not detected with shotgun proteomics in enriched samples	no	0
4WPI	icp0, USP7	redundant info to 4WPH	no	0
4ZXS	UL31, UL34	interlinks only in N-terminal disordered region of NEC2, which is not in structure	yes	0
5C56	icp0, USP7	redundant info to 4WPH	no	0
5U1D	US12, TAP1, TAP2	in Figure 5	yes	1

5XO2	gB, PILRA	PPI with cell surface receptor for entry	no	0
6CGR	MCP, SCP, TRX1, TRX2, CVC1, CVC2, UL36	No apparent reason for not identifying cross-links	no	0
6FAD	icp27, srpk1	only 16 residues of icp27 in structure with no Lysine	no	0
6JXU	icp0, sumo	20 residues of icp0 in structure, no lysine	no	0
6JXV	icp0, sumo	redundant info to 6JXU	no	0
6JXW	icp0, sumo	redundant info to 6JXU	no	0
6JXX	icp0, sumo	redundant info to 6JXU	no	0
6M5R	TRM1, TRM3, TRM2	TRM1 not detected with shotgun proteomics in enriched samples TRM3 detected only in 1 out of 4 replicates with shotgun proteomics in enriched samples	no	0
6M5S	TRM1, TRM3, TRM2	redundant info to 6M5R	no	0
6M5U	TRM1, TRM3, TRM2	redundant info to 6M5R	no	0
6M5V	TRM1, TRM3, TRM2	redundant info to 6M5R	no	0
6ODM	MCP, SCP, TRX1, TRX2, CVC1, CVC2, UL36	Redundant to 6CGR	no	0
6T5A	UL51, UL7	No apparent reason for not identifying cross-links	no	0
8EXX	UL30, UL42	redundant info as in 9ENP Fig 5	yes	2
8G6D	UL31, UL34	Redundant to 4ZXS	yes	0
8KFA	gB, D48 Fab	Interaction with antibody	no	0
8OJ6	UL30, UL42	redundant info as in 9ENP Fig 5	no	2
8OJ7	UL30, UL42	redundant info as in 9ENP Fig 5	no	2
8OJA	UL30, UL42	redundant info as in 9ENP Fig 5	no	2
8RGZ	gB, HDIT101 Fab	Interaction with antibody	no	0
8RH0	gB, HDIT102 Fab	Interaction with antibody	no	0

8V1Q	UL30, UL42	redundant info as in 9ENP Fig 5	yes	2
8V1R	UL30, UL42	redundant info as in 9ENP Fig 5	yes	2
8V1S	UL30, UL42	redundant info as in 9ENP Fig 5	yes	2
8V1T	UL30, UL42	redundant info as in 9ENP Fig 5	yes	2
9ENP	UL30, UL42	2 links in Fig 5	yes	2

We now mention the problem that only few virus-derived multimeric structures exist in the introduction section of our revised manuscript:

“While structural biology techniques have provided insight into the assembly of various viral protein complexes¹⁻⁵, only a limited number of virus-host structures has been reported.”

To reflect on this number, we changed the sentence the referee referenced in the Results section to:

“Only 5 inter-links (3 PPIs) could be mapped onto existing experimental structures of heterodimers (Figure 5a), reflecting the paucity of structural information on multimeric HSV-1 complexes for which to date only ~11 complexes have been solved outside of receptor or antibody binding.”

(16)- On lines 339-341 the authors state that “We obtained 68 models (18 involving viral proteins) that have both acceptable docking scores (pDockQ > 0.23) and an agreement of at least half of the inter-links with the prediction (Supplementary Table 4).” Is not the agreement of only half of the observed inter-links with the prediction a quite low agreement? Does the crosslinking data really support the structural models under these conditions? How was the crosslinking data filtered (for reliable hits)?

We chose the filtering criteria as at least half of the cross-links in agreement with the model based on our recent study on system-wide AlphaFold analysis of a ~2,000 protein-protein interactome from HEK293T cells (Bogdanow et al., Mol Syst Biol 2024, see: [10.1038/s44320-024-00079-w](https://doi.org/10.1038/s44320-024-00079-w)). Figure 5 in that publication shows the mapping analysis of cross-links onto AlphaFold structures, with panels c and d showing which fraction of cross-links satisfies the distance constraint. The cross-link distribution shows that for acceptable models, we can expect at least 50 % agreement, while in good models this can be 60-70 %. e decided to take a lenient threshold (>50%) to retain all acceptable models, which give us a broader range of selection for biochemical validation by e.g., mutagenesis of the binding interfaces. As an example, mutational analysis validated the DDB1-UL47 interaction, which had a 50% agreement of the cross-links with the model.

Please also note that, following the revision of the used AF score, as requested by reviewer#1 - point#2, we obtained 61 models (12 viral) above model confidence score of 0.5 and 50 % XL satisfaction rate.

Commenting on the reliability of the data, we mentioned in our response to reviewer #2, point 6/7 the specifics of FDR control in our study, which was performed at a 1 % FDR, separately for intra- and inter-links. This is now standard in the XL-MS field and accurately reflects error rates in ground-truth data (Bogdanow et al., Mol Syst Biol, 2024, <https://doi.org/10.1038/s44320-024-00079-w>).

(17)- The authors apply several different proteomics methods in their paper to address the HSV-1 – host interactome including SHVIP (i.e. labeling, XL-MS, and enrichment), AP-MS, co-IP and SILAC labeling. An overview presentation of these workflows and how they are connected in generating the various results and contributing to the different conclusions made in the manuscript would increase the understanding of the methodology for the broader audience.

We have included method schemes upon first appearance of each proteomic method in the manuscript: This includes SHVIP (XL-MS + labeling, Figure 1a), SILAC AP-MS (Supplementary Figure 4b) and label-free AP-MS (Supplementary Figure 5e & Figure 4a). For additional aid we now include Supplementary Table 7 summarizing which experimental design contributed to which figure panel.

(18)- On lines 351-352 the authors state that “Two UL47-DDB1 cross-links were identified, with one of them satisfying the distance constraint in the corresponding AF2 model.” which is also highlighted in Figure 5b. There are only two crosslinks identified, one that supports the AF2 model, and one that does not. Can the authors comment on the reliability of the data? Is there any indication of flexibility in the regions involving the violated crosslink?

There are multiple conceivable explanations for this disagreement but they are all hypothetical:

- *The overall flexibility or different multimerization states of UL47-DDB1, which are common features for herpesvirus tegument proteins. The violated cross-link may represent one of these states.*
- *The UL47 gene is known to generate two protein products, which are also known as VP13 and VP14 (MacLean et al., J Gen Virol, 1990 [10.1099/0022-1317-71-12-2953](https://doi.org/10.1099/0022-1317-71-12-2953)), which are created based on an unknown mechanism. The violated cross-link may stem from the alternative protein product.*
- *The association of UL47 with DDB1 likely results in a catalytic reaction, as DDB1 is part of a CRL4 E3 ubiquitin ligase that typically marks certain protein substrates for degradation. The catalytic process may indeed require some flexibility, which may have been captured by the violated cross-link.*

In the end, the structural model is “only” one possible snapshot of this interaction, and to draw more concrete conclusions, we need additional evidence, such as mutational analysis as we performed in this study (Figure 6b-d).

Regarding reliability, please see our response one of the previous points (#16).

(19)- The authors state on lines 461-462 that “Augmenting AF2 with SHVIP strengthened 18 protein complex predictions, which are potentially linked to DNA replication, immune evasion, and

signaling.” I emphasize here the word “strengthened”. How many of these interactions/complexes are novel, i.e. identified based on the data presented in this manuscript, and how many interactions (not structure of complexes) were previously known?

With “strengthen” we intended to emphasize the advantage of adding residue-level cross-link data as a way to cross-validate computational models with orthogonal experimental data. This is especially important in our case, since host-virus and virus-virus dimers generally have lower AlphaFold confidence scores than host-host interactions. Using cross-link agreement as an additional filtering criterion gives us the opportunity to also consider dimeric predictions with lower scores that nevertheless agree with the cross-link restraints. We changed the sentence to:

“Augmenting AF2 with SHVIP proved to be effective to cross-validate generally lower-scoring host-virus protein complex predictions and thus identify convincing structures, which are in this case potentially linked to DNA replication, immune evasion, and signaling.”

According to this comment and also to the reviewer’s 2nd major comment (point #12) above we now also modified Supplementary table 2 including a table summarizing which of the detected PPIs had previous evidence in the literature.

Minor comments:

(20)- The abbreviation (SHVIP) of the established workflow is not mentioned in the abstract and should be included there.

We added the abbreviation SHVIP to the abstract.

(21)- On lines 52-54, the authors state that “However, AF has been developed mainly based on structural data of multimeric complexes within one organism, with co-evolutionary constraints likely different in host-pathogen scenarios.” References should be cited to support this statement.

We added the references below:

<https://doi.org/10.1146/annurev-genom-120622-020615>

<https://doi.org/10.1016/j.sbi.2023.102529>

(22)- The reference to “Both challenges...” on line 77 is unclear; does it reflect the challenges presented on line 66-76 or the broader topics presented in the paragraphs on lines 43-57 (structural context and molecular determinants) or 58-76 (molecular environment). Please rephrase to improve readability and clarity.

We replaced “Both challenges” with “These challenges”.

(23)- Please include more info and detail on the methodological setup for SHVIP, presented currently in Figure 1A, and please increase the image size/font size.

We updated the figure according to the reviewer's comment. We re-arranged panels and extended labels to facilitate a better understanding. However, we did not go into high detail on the methodological aspects, as we present SHVIP as a generally applicable method for different viruses/cell culture systems for which the method details like infection time points or concentrations of cross-linker/amino acid analog may differ. To facilitate the accessibility of the figure we increased font size and moved the network from Figure 1d to a dedicated new Figure 2 in the revised version of the manuscript.

(24)- Please verify that the text passage on lines 133-135 “Both the relative frequency and absolute numbers of PPIs involving viral proteins increased upon HPG enrichment (Supplementary Figure 1c) and showed reproducibility similar to previous XL-MS studies from complex samples 36 (Supplementary Figure 1d).” refers to the correct figure. Currently the Supplementary Figure 1c and d represent a comparison between Exp 1 and Exp2; not data from previous studies. Also, reading from the beginning of the manuscript Exp1 and Exp2 have not been described prior to this text passage.

Thanks for pointing this out. Accordingly, we added an introduction on both the acquisition schemes and moved the reference to Supplementary Figure 1d to the middle of the sentence. This makes it clearer that the reference refers to published data and replicate comparisons within this other work.

*“We evaluated two acquisition schemes, based on either MS2-MS3 (Experiment 1) or MS2-only (Experiment 2) fragmentation. Both the relative frequency and absolute numbers of PPIs involving viral proteins increased upon HPG enrichment in both experiments (**Supplementary Figure 1c**). Reproducibility between the two experiments (**Supplementary Figure 1d**) was similar to previous XL-MS studies from complex samples³⁶.”*

(25)- Please increase the image size and font size of Figure 1d. The edges connecting the nodes are very thin, and it difficult to verify that all human host proteins presented in the figure are in fact primary or secondary interactors. Also, black font on dark blue/purple is hard to see. Here, in this figure the HSV-1 proteins are mentioned, but there is not a more general presentation of the virus (please see major concern above) for the reader to connect these in terms of viral biology.

Following the reviewer's comment, we reformatted Figure 1d, which is now Figure 2a. Since the network now occupies a larger space, we also included host protein labels. To increase contrast and visibility, we increased the thickness of connections and the paleness of viral nodes. To provide the reader with some orientation on the nature of viral proteins included in the network, we indicated their belonging to the four main HSV-1 protein classes (glycoproteins, tegument, capsid or non-structural proteins) by colored node outlines. We further included, as suggested, a schematic illustration of the virus particle in Figure 2a and used the same color code to allocate the viral protein classes to the different virion layers.

(26)- Please clarify what the difference is between Figure 1d and Supplementary Figure 2a. Is the only difference that the human proteins are indicated in the latter? If so, please merge into one single figure only and increase the node font size.

Following the reviewer's suggestions, we now merged all information from Figure 1d and Supplementary Figure 2 of the original manuscript into the new Figure 2 of the revised manuscript - see also our response to point (23).

(27)- On lines 151-158 the authors describe that they combine data from two MS (MS2 and MS2/MS3) experiments to provide a high-coverage structural interactome of the infected cell. Please elaborate on the rationale for using the two methods in benefit of the broader audience; I assume that this is due to DSSO being MS-cleavable and the gain in data depth of combining these two approaches.

The reviewer is right. We performed these two experiments primarily to provide evidence of the performance of the workflow when using different acquisition schemes.

We added the following sentence to the methods section when introducing these two experiments:

"We conducted two experiments with different MS acquisition schemes (MS2-MS3, and MS2 only) to provide evidence of the performance of the workflow when using different acquisition schemes."

(28)- Line 213-215, which are the five ER membrane proteins connected to the NEC?

We mention the five ER membrane proteins IKBIP, SEC22B, SSR3, SEC61B and LRRC59 now in the text.

(29)- The line thickness between the two groups "XL within compartment" and "XL connecting compartments" are identical and cannot be discerned.

We adjusted line thickness and color. It is now easier to discriminate between the two categories of cross-links (within and between compartments).

(30)- Figure 4f does not in its current format show the violated crosslinks (>35 Å) (as indicated in the figure legend key).

There are no violated cross-links for these structures. To emphasize this we mention this in the figure legend and also removed the violating category in the panel.

(31)- In Supplemental Figure 5a and Supplemental Figure 6a-c the contrast (choice of colors) makes it difficult to distinguish the residues highlighted in magenta. This choice of colors works better in Figure 6e where the residues in magenta are indicated in bolder lines.

We adjusted the thickness of the cartoon depiction for the loops, so that the loops are now better visible.

Reviewer #4 (Remarks to the Author):

We thank the reviewer for her/his efforts for training in peer review and support appropriate recognition for early career researchers.

We thank all three reviewer's again for providing valuable feedback on the revised version of our manuscript. We were glad to read that we have addressed most of the concerns. Below, we respond point-by-point to the remaining concerns.

REVIEWER COMMENTS

Reviewer #1 (Remarks to the Author):

I appreciate the authors' efforts in addressing my comments. The paper is now in a more technically sound state in terms of modeling. However, I still have one major remaining concern regarding the AlphaFold version used in this study.

The authors made a notable effort to run AF2.3. However, I do not understand why they did not use AF3, as they should have had access to its stand-alone version during the revision period. AF3 runs significantly faster than AF2.3, and this paper could have provided a valuable platform for comparing its results with those of AF2.2 and AF2.3.

Additionally, AF2.3 is only a slightly improved version of AF2.2. The results would have differed more significantly if the authors had used AF2.1 (and 2.3 is not necessarily better than 2.1). In this regard, I recommend that the authors consult the following papers:

- <https://pubmed.ncbi.nlm.nih.gov/37548092/>

- <https://pubmed.ncbi.nlm.nih.gov/39528570/>

Given the innovative nature of this work, I believe it should align with the latest modeling technologies available in the field. Therefore, if the editor agrees, I suggest another round of revision, at the very least incorporating results from AF3.

We thank the reviewer for her/his overall positive comments. We agree that testing the newest AlphaFold version in addition to our previous analysis with AlphaFold 2.3 is reasonable. We therefore predicted all host-host, virus-host, and virus-virus heterodimers in our network with AlphaFold 3 and compared AlphaFold metrics and agreements with cross-linking distance constraints.

We find that, in general, higher-scoring predictions agree better with the cross-linking data than lower-scoring predictions (Reviewer Figure 1a). When comparing the AF2.3 and AF3 versions, we find that the scoring of many matches agrees between both versions. However, AF2.3 has a general trend for slightly higher scores than AF3. This is observed for all kinds of PPIs in our manuscript, host-host, host-virus, virus-virus (Reviewer Figure 1b). However, the differences in scores probably do not reflect differences in prediction accuracy, as the ipTM scores are calculated differently for both AF versions.

Reviewer Figure 1: Evaluation of AlphaFold3 performance on SHVIP heterodimers. (a) agreement of inter-link distance with heteromeric models, binned by confidence scores. The DDSO distance criteria of 35 Å is indicated with a line. **(b)** Scatterplot of confidence scores by AlphaFold2 and AlphaFold3 for the same dimeric models. Dimers containing viral proteins are labeled green.

We also closely evaluated all predictions using the criteria we outlined for AF2.3 in our manuscript. While overall fewer models could meet these criteria with AF3, due to the lower *ip*TM and *p*TM scores, some additional predictions with AF3 now meet our criteria. Focusing on virus-host interactions, this includes the heterodimer between STT3A, a part of the N-oligosaccharyltransferase complex, and glycoprotein B, a viral envelope protein (Reviewer Figure 2a), and UL45 membrane protein with peptidyl-prolyl isomerase B (PPIB) (Reviewer Figure 2b).

Reviewer Figure 2: AlphaFold3 delivers novel heterodimeric models partially in agreement with the cross-linking data. (a) Dimer of the viral glycoprotein gB and host STT3A, and (b) dimer of viral envelope protein UL45 and host PPIB. Confidence scores and detected interlinks are shown (inter-links >35 Å blue, >35 Å red). PAE (predicted alignment error) plots are depicted below the models.

We decided to include Reviewer Figure 1a, 2a-b as new Supplementary Figure 4. Additionally, we performed an analysis of confidence scores by type of dimeric prediction and an analysis of the pLDDT score at the XL-site, included in Supplementary Figure 4. We mention the set of AF3 predictions in the following section within the results:

“To test whether SHVIP is compatible with newer structure modeling tools, we additionally tested its performance using AlphaFold 3 (AF3), with overall similar performance to AF2 (Supplementary Figure 4a-c). In general, higher scoring predictions aligned more closely with the distance constraints than lower scoring ones (Supplementary Figure 4d). Applying the same model selection criteria across both confidence categories (see above), we identified two additional models that matched at

least 50% of the detected inter-links from structured regions. These include STT3A, the catalytic subunit of the ER-resident oligosaccharyltransferase complex, known to be crucial for HSV-1 glycosylation⁷³, with glycoprotein B, and the viral membrane protein UL45 in complex with PPIB (Supplementary Figure 4e). These findings demonstrate that SHVIP is also compatible with AF3, which may outperform AF2 for certain targets.”

We thank the reviewer for the suggestion and feel that including AF3 prediction has improved the quality of our manuscript.

In addition, the reviewer asked to reflect on massive sampling strategies for our pipeline. We agree with the reviewer that this is an interesting idea. While massive sampling strategies can indeed produce better models, the problem appears to be that they are computationally costly, requiring thousands of predictions per candidate. As we have more than 700 pairs, we would be required to run millions of predictions, which we think is outside the scope of our manuscript at this stage. However, we think that it is indeed interesting to consider this option in the future, and therefore added a sentence in the discussion section:

“To arrive at even more confident predictions, it will be promising to improve structure modeling, as e.g. through approaches that incorporate cross-linking distance constraints⁸⁷ or massive sampling strategies^{88,89} in the future.”

Reviewer #2 (Remarks to the Author):

Based on the authors’ revised manuscript and detailed response, all of this reviewer’s initial concerns regarding the technical aspects of mass spectrometry analysis and specificity have been satisfactorily addressed. The authors have provided clear and comprehensive explanations regarding the specificity of their data, covering both the chemical and biological aspects of the cross-linking approach. The methodological rigor and the resulting dataset represent a significant advancement in the structural characterization of virus-host protein-protein interactions.

However, the authors did not adequately address Comment 1 from this reviewer. Demonstrating at least one instance of biological significance—specifically, a clear contribution to viral replication and/or pathogenesis—of one of the newly identified HSV-1–host interactions during viral infection would be important to support the study’s validity. Unfortunately, the interaction between UL47 and UL48 presented in the Response to Referees Letter does not represent a newly identified HSV-1–host interaction but rather an HSV-1–HSV-1 interaction. Furthermore, this interaction is likely an unsuitable model for a novel interaction, as the intracellular binding of UL47 to UL48 has already been reported (PMID: 16014918) and the binding in HSV-2–infected cells has also been documented (PMID: 11087097). Additionally, a critical issue remains: the effects of the Δ 2-15 and Δ 469-486 mutations in UL48, which disrupt the binding between UL47 and UL48, are more pronounced than those observed with a null mutation in UL47. It has been reported that the deletion mutation in UL47 does not affect the expression level of UL48 (PMID: 8382306), nor does it reduce HSV-1 replication by more than 2 logs. Moreover, the series of experiments lacks appropriate controls (e.g., a repair virus).

Given these limitation, it remains questionable whether this methodology alone is not sufficient for publication in this journal.

We thank the reviewer for the careful and thorough analysis of the preliminary findings that we included in the last rebuttal letter.

The reviewer is correct, UL47 and UL48 are both viral proteins, and an interaction (i.e., binding) in a large-scale Yeast-two-hybrid assay has been previously observed. However, the reference given by the reviewer on UL47-UL48 binding in HSV-2 infected cells considers UL46 and not UL47. The reviewer is also perfectly correct in pointing out the critical issue of UL47 deletion mutants that do not mimic the phenotype of our $\Delta 2-15$ and $\Delta 469-486$ mutants.

For this and the other reasons we mentioned in our first revision round, we judged our data with $\Delta 2-15$ and $\Delta 469-486$ mutants as too preliminary to include them in the manuscript (see our first point-by-point response) and agree that thoroughly assessing the role of UL47-UL48 binding requires substantially more experiments. For this reason, the data is on UL47-UL48 binding and mutagenesis is not included in our manuscript and will be thoroughly assessed in an independent publication.

Regardless of this issue, we think that our manuscripts has two major points that justify a publication in Nature communications: (i) the method – the first method that allows characterizing the native virus-host structural interface at proteome-scale from actively replicating virus, which is also applicable to other viruses (please see our recent work in collaboration with Kosinski lab using the same methodology for Influenza A virus: <https://www.biorxiv.org/content/10.1101/2025.03.09.642134v2>) and (ii) as a resource – We present the first system-wide analysis of the structural virus-host interface of any virus. We regard our data as very reliable, due to the extensive coverage, physiological relevance and rigorous validation, including orthogonal methods and mutational analyses.

Reviewer #3 (Remarks to the Author):

Bogdanow et al. describe the development of deep in-cell virus-host protein-protein interaction (PPI) profiling by combining proteome-wide cross-linking mass spectrometry (XL-MS) with selective enrichment of newly synthesized viral proteins; Structural Host Virus Interactome Profiling or SHVIP.

I have carefully assessed the revised manuscript. The authors have addressed all my concerns and comments, and I have no further remarks on the manuscript.

Thank you!

Reviewer #4 (Remarks to the Author):

Thank you!